# Proteo-genomic characterization of virus-associated liver cancers reveals potential subtypes and therapeutic targets

Masashi Fujita [1], Mei-Ju May Chen[2], Doris Rieko Siwak[3], Shota Sasagawa[1], Ayako Oosawa-Tatsuguchi[1], Koji Arihiro[4], Atsushi Ono[5], Ryoichi Miura[5], Kazuhiro Maejima[1], Hiroshi Aikata[5], Masaki Ueno[6], Shinya Hayami[6], Hiroki Yamaue[6], Kazuaki Chayama [5], Ju-Seog Lee [3], Yiling Lu[3], Gordon B. Mills [7], Han Liang [2,3], Satoshi S. Nishizuka [8] & Hidewaki Nakagawa [1] ✉

Primary liver cancer is a heterogeneous disease in terms of its etiology, histology, and therapeutic response. Concurrent proteomic and genomic characterization of a large set of clinical liver cancer samples can help elucidate the molecular basis of heterogeneity and thus serve as a valuable resource for personalized liver cancer treatment. In this study, we perform proteomic profiling of ~300 proteins on 259 primary liver cancer tissues with reverse-phase protein arrays, mutational analysis using whole genome sequencing and transcriptional analysis with RNA-Seq. Patients are of Japanese ethnic background and mainly HBV or HCV positive, providing insight into this important liver cancer subtype. Unsupervised classification of tumors based on protein expression profiles reveal three proteomic subclasses R1, R2, and R3. The R1 subclass is immunologically hot and demonstrated a good prognosis. R2 contains advanced proliferative tumor with *TP53* mutations, high expression of VEGF receptor 2 and the worst prognosis. R3 is enriched with *CTNNB1* mutations and elevated mTOR signaling pathway activity. Twenty-two proteins, including CDK1 and CDKN2A, are identified as potential prognostic markers. The proteomic classification presented in this study can help guide therapeutic decision making for liver cancer treatment.

Primary liver cancer is a highly heterogeneous disease. Its histology is categorized into hepatocellular carcinoma (HCC), intrahepatic cholangiocarcinoma (ICC), and cHCC-ICC (combined hepatocellular carcinoma-intrahepatic cholangiocarcinoma) in order of decreasing frequency[1]. In Asian countries where HCC is more prevalent, hepatitis B virus (HBV) or hepatitis C virus (HCV) represent the dominant drivers[2]. In western countries, alcoholic liver disease and nonalcoholic steatohepatitis or metabolic diseases are more prevalent drivers[3]. First-line

[1]Laboratory for Cancer Genomics, RIKEN Center for Integrative Medical Sciences, Yokohama, Japan. [2]Department of Bioinformatics and Computational Biology, The University of Texas MD Anderson Cancer Center, Houston, TX, USA. [3]Department of Systems Biology to Genomic Medicine, The University of Texas MD Anderson Cancer Center, Houston, TX, USA. [4]Department of Anatomical Pathology, Hiroshima University Hospital, Hiroshima, Japan. [5]Department of Gastroenterology and Metabolism, Graduate School of Biomedical and Health Sciences, Hiroshima University, Hiroshima, Japan. [6]Second Department of Surgery, Wakayama Medical University, Wakayama, Japan. [7]Division of Oncological Sciences, Knight Cancer Institute, Oregon Health & Science University, Portland, OR, USA. [8]Division of Biomedical Research and Development, Institute of Biomedical Sciences, Iwate Medical University, Morioka, Japan. ✉e-mail: hidewaki@riken.jp

combination therapy with atezolizumab and bevacizumab has demonstrated exciting activity in advanced HCC, however, the response rate is only about 30% and patients almost inevitably progress[4]. The molecular basis of this variable response and predictive markers able to identify patients likely to obtain benefit remain to be identified.

Proteins are the primary functional unit of the cell. RNA and protein levels are only weakly correlated particularly for post-transcriptional programs such as regulatory phosphorylation, limiting the utility of transcriptional profiling[5,6]. Proteomic profiling of a large number of clinical tumor tissues thus has the potential to provide some insights into the molecular heterogeneity of primary liver cancer. Three studies have analyzed more than one hundred liver cancers using mass spectrometry-based global proteomics. Jiang et al. analyzed 110 early-stage HCC and identified, on average, 5953 proteins per tumor[7]. Gao et al. examined HBV-related HCC from 159 patients, detecting, on average, 8934 proteins per sample[8]. Dong et al. performed proteomic profiling on 214 ICC and found 7385 proteins per sample[9]. Although these studies provided an unbiased view of liver cancer proteome, their sample sizes and quantitative analyses were still limited. Reverse-phase protein array (RPPA) provide a high-throughput, cost-effective technology for measuring protein expression levels on a large number of samples[10]. Hundreds of tissue lysates are spotted on a slide, and quantification of protein abundance is performed by applying antibodies to the slide. Even if tissue samples have a small volume, a curated and well-validated panel of antibodies allows sensitive measurement of protein abundance. RPPA is particularly applicable to analysis of post-translational modification such as protein phosphorylation[11]. RPPA has been already applied to molecular profiles of many types of cancers including 184 HCC as part of The Cancer Genome Atlas (TCGA) project[12]. However, their study was limited not only in the sample size but also by diversity in patient's ethnicity and disease etiology.

In this work, we report a proteomic profiling of ~300 proteins on 259 primary liver cancers, mostly positive for HCV and HBV, with RPPA and provide insight into potential subtypes and therapeutic strategies for liver cancer.

## Results

### Protein abundance correlated with mRNA abundance

We first examined correlation between protein abundance and mRNA abundance in tumor tissues. Protein abundance was generally positively correlated with mRNA abundance albeit with a relatively low correlation (Fig. 1A). The correlation coefficient of 0.25 was significantly higher than that of randomly shuffled pairs ($p < 2.2 \times 10^{-16}$ by Student's $t$-test). Among 222 mRNA-protein pairs, 30 pairs (14%) had negative correlation coefficients. There are multiple potential reasons for the low correlation with these proteins including low-expression levels, low dynamic range, tissue heterogeneity, and post-transcriptional regulation[13]. As expected, there was a much weaker correlation between phosphoprotein abundance and corresponding total protein abundance emphasizing the utility of the RPPA assay (Fig. 1B). The mean correlation coefficients among 40 total-phosphoprotein pairs was 0.06. Of the 40 pairs, 18 pairs (45%) had negative correlation coefficients.

We examined whether mRNA-protein pairs have a similar level of correlations in cancer cohorts of TCGA (Supplementary Fig. 2). There were significant similarities between mRNA-protein correlations in this study and those in TCGA. As expected, the TCGA liver cancer cohort (LIHC) had the highest similarity with our liver cancer cohort (Spearman's correlation r = 0.62).

### Local and distant interaction between somatic copy number alterations and protein abundance

Somatic copy number alterations (SCNAs) may impact both mRNA and protein abundance. We examined 209 mRNA-protein pairs for their association with SCNA in tumor. Abundance of 108 mRNA (52%) had significantly positive correlation (FDR < 0.05) with SCNA at the same chromosomal locus, indicating local interaction between them (Fig. 2A). Moreover, we observed numerous distant interaction between mRNA and SCNA on different chromosomes. The SCNAs of 4q, 16q, 17p, and 5q had the largest number of distant interaction with mRNA abundance. Protein abundance had sparse interaction with SCNA both locally and distantly (Fig. 2B). Only 30 of 209 proteins (14%) had significant local interaction with SCNA including 1p, 1q, 4q, 5q, 9, 10, and 22 that were observed in the RNA local interactions. These included correlations with SCNAs of the distant interaction of 4q, 17p, and 5q observed with RNA remained prominent in the protein SCNA correlations. Loss of 17p was observed in 49% of cases, and the chromosome arm has a tumor suppressor gene TP53. Loss of 4q was found in 35% of cases, and the arm contains interferon regulatory factor 2 (IRF2) gene. A previous study reported that IRF2 is a tumor suppressor gene as its inactivation impairs TP53 function in HCC[14]. Gain of 5q was found in 25% of cases, and the arm contains FGFR4 gene. Nevertheless, these data indicate that the effects of SCNA on protein levels are less than on RNA levels as expected by the relatively low correlation between RNA and protein levels noted above (Fig. 1A).

**Somatic alterations of driver genes correlated with their protein abundance.** We compared somatic mutations of two driver genes, CTNNB1 and TP53, with their protein expression levels. Mutations of CTNNB1 were associated with elevated protein level ($p = 0.0033$) (Fig. 2C). Mutations of CTNNB1 disrupt phosphorylation sites of its exon 3, thereby preventing degradation of the protein[15]. The elevated CTNNB1 protein level is consistent with this mechanism. On the contrary, mutations of TP53 were significantly associated with decreased protein level ($p = 0.029$) (Fig. 2D). This may be explained by frequent copy number loss of TP53. Indeed, in our dataset, 88% of TP53-altered tumors had copy number loss of TP53.

### Proteomic subclasses of liver cancer

To obtain insights into heterogeneity of clinical liver cancer, we performed hierarchical clustering of 259 tumors based on protein abundance. A consensus clustering algorithm identified three subclasses of tumors that were stable to random subsampling (Supplementary Fig. 1). We named the three subclasses R1, R2, and R3 (Fig. 3A). Of 259 tumors, 53, 100, and 106 tumors were classified to R1, R2, and R3, respectively. The proteomic subclasses had various associations with clinical features. The R1 subclass was younger (65 years in R1 vs 70

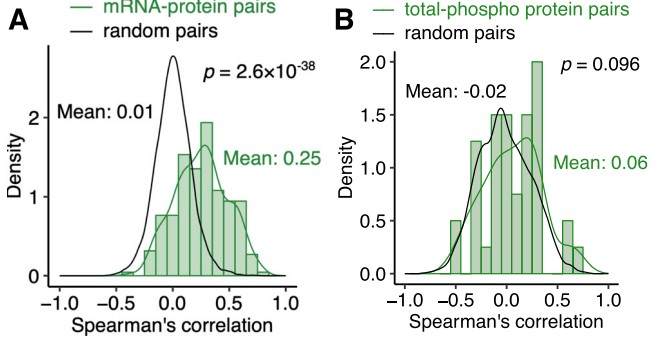

**Fig. 1 | Correlation between expression levels of mRNA, protein, and phopho-proteins in 259 primary liver cancers. A** Spearman's correlation of expression levels in 222 mRNA-protein pairs are shown in green. Correlation in 22,200 randomly shuffled pairs are shown in black. **B** Correlation in 40 pairs of total and phosphorylated proteins are shown in green. Correlations in 4000 random pairs are shown in black. Statistical significance was assessed using two-sided Student's $t$-test. Source data are provided as a Source Data file.

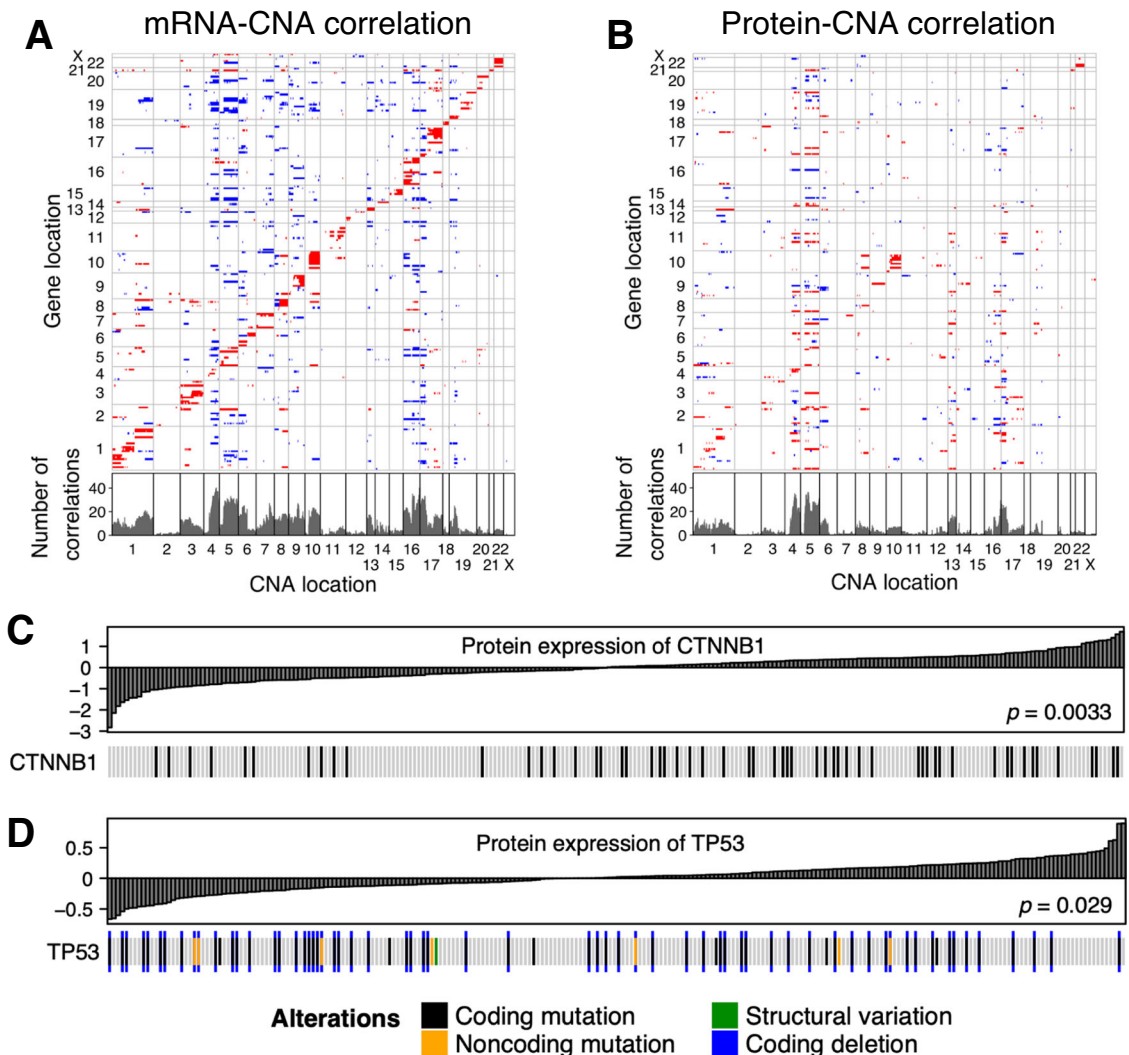

**Fig. 2 | Correlation between protein expression levels and somatic alterations. A**, **B** Local and distant effects of somatic copy number alterations on gene expression levels in tumor. Correlation of CNA with **A** mRNA expression levels and **B** protein expression levels. The horizontal axis is the genomic coordinates of CNAs. The vertical axis represents the genomic coordinates of 209 genes, which were identical between mRNA and protein analysis. Spearman's correlation between copy number signals and expression levels were computed. Significant correlations (FDR < 0.05) were depicted either red for positive correlations or blue for negative correlations. Bottom panel shows the number of significant correlations for each CNA locus. **C**, **D** Effects of somatic driver alterations on expression levels of their protein product. Tumors were sorted by protein expression level of **C** CTNNB1 and **D** TP53. The *p*-values were computed by two-sided Wilcoxon rank sum test comparing protein expression levels between wild-type and altered tumors. Source data are provided as a Source Data file.

years in non-R1, median; *p* = 0.010 by Wilcoxon rank sum test), more likely to be HBV-positive (40% vs 22%; *p* = 0.013 by Fisher's exact test), and to have non-HCC histology (ICC and cHCC-ICC in R1, 15 of 53 tumors (28%); ICC and cHCC-ICC in R2 and R3, 11 of 206 tumors (5%); *p* = 1.0 × 10⁻⁵). The R2 subclass had larger tumor size (36 mm vs 28 mm, median; *p* = 0.0037), vascular invasion (45% vs 26%; *p* = 0.0029), advanced tumor stage (stage IV; 16% vs 5%; *p* = 0.0040), higher serum α-fetoprotein (AFP) (>200 ng/ml; 43% vs 18%; *p* = 2.9 × 10⁻⁵), and more likely to be HCV-positive (65% vs 52%; *p* = 0.039). The R3 subclass showed a weak trend toward NBNC (negative for HBV and HCV; 25% vs 16%; *p* = 0.081 by Fisher's exact test).

Proteins selectively upregulated in the R1 subclass included annexin A1 (ANXA1), WW domain containing transcription regulator 1 (WWTR1), β-actin (ACTB), spleen associated tyrosine kinase (SYK), and pyruvate kinase M1/2 (PKM) (Supplementary Fig. 3). The R2 subclass was enriched for phosphoribosylaminoimidazole carboxylase and phosphoribosylaminoimidazole-succinocarboxamide synthase (PAICS), Ras homolog, mTORC1 binding (RHEB), Parkinsonism associated deglycase (PARK7), ERBB receptor feedback inhibitor 1 (ERRFI1),

and Janus kinase 2 (JAK2). The R3 subclass was characterized by elevated protein expression of progesterone receptor (PGR), unc-51 like autophagy activating kinase 1 (ULK1-pSer757), dual specificity phosphatase 4 (DUSP4), AKT serine/threonine kinase 1 (AKT1-pSer473), and WEE1 G2 checkpoint kinase (WEE1). These proteins may serve as biomarkers for identifying the proteomic subclasses using immunohistochemistry.

Pathway activity score was computed based on protein abundance (Fig. 3B). Apoptosis pathway was selectively activated in the R2 subclass (*p* = 4.1 × 10⁻¹⁶ by Wilcoxon rank sum test). TSC/mTOR, receptor tyrosine kinase (RTK) pathway, and hormone signaling pathways were selectively activated in the R3 subclass (*p*-values < 0.0022). The R2 subclass was associated with poor overall survival compared to the other two subclasses (*p* = 0.0008 by log-rank test) (Fig. 3C). Disease-free survival was comparable among the proteomic subclasses (*p* = 0.29).

To examine the robustness of our classification, we applied 10-fold cross-validation to the protein expression matrix of this Japanese cohort (Fig. 3A; 122 proteins × 259 tumors). Tumors were randomly

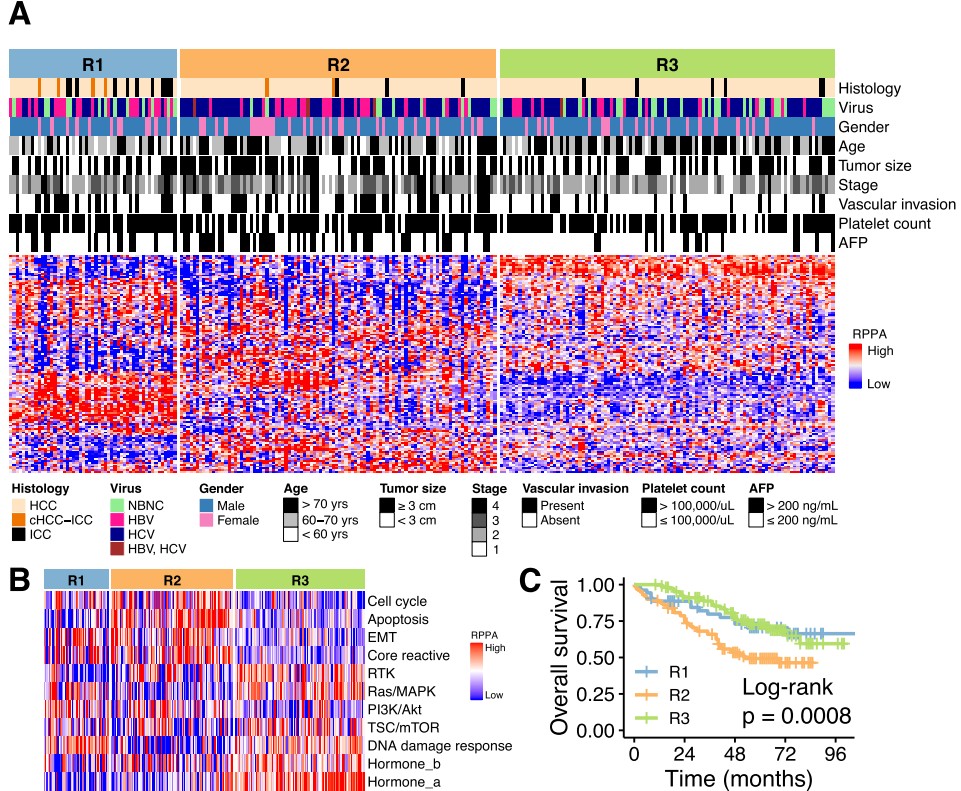

**Fig. 3 | Proteomic subclasses of primary liver cancers. A** Proteomic subclasses R1–R3, clinical features, and protein expression levels in 259 tumors. Heatmap shows expression levels of 122 proteins that were used for consensus clustering of the tumors. HCC hepatocellular carcinoma, ICC intrahepatic cholangiocarcinoma, cHCC-ICC combined HCC-ICC, NBNC non-hepatitis B non-hepatitis C. **B** Pathway scores and the proteomic subclasses in 259 tumors. **C** Kaplan–Meier plots of overall survival after surgical treatment. The number of patients were 53 in R1, 100 in R2, and 106 in R3. The *p*-value was computed by the log-rank test. Source data are provided as a Source Data file.

split into 10 subsets. Nine of them were used for training, the remaining one was used for testing, and this process was repeated 10 times to compute the average prediction accuracy. Naive Bayes classifier was trained to predict subclasses R1, R2, and R3 from protein expression levels. Its average prediction accuracy was high (91.5%). When *k*-nearest neighbor classifier was used as a prediction model, it also showed high prediction accuracy (91.1%; $k = 3$).

To validate these observations in an independent cohort, we examined RPPA data of HCC in the TCGA project. Of 162 TCGA tumors, 21, 46, and 95 tumors were assigned to R1, R2, and R3 subclasses, respectively (Supplementary Fig. 4A). The R2 subclass had more vascular invasion (58% vs 23%; $p = 0.0003$) and stage II or higher tumors (79% vs 50%; $p = 0.0010$). But there was no association between overall survival and subclasses in TCGA ($p = 0.3$), which may be related with the differences of etiology and treatment strategies for liver cancers between Japan and USA.

### Genomic and transcriptomic features of the proteomic subclasses

To identify potential somatic driver alterations of the liver cancer samples, we analyzed whole-genome sequencing data. Among the identified events, *TP53* and *CTNNB1* mutations were enriched in the R2 and R3 proteomic subclasses, respectively (FDR < 0.05) (Fig. 4A).

Previous studies have explored transcriptome data and proposed molecular classifications of liver cancer. We compared our proteomic subclasses with previous transcriptomic classifications to examine their correlations (Fig. 4B). Because the transcriptomic classification has been primarily focused on HCC, the comparison was performed using only HCC in our sample set. The proteomic subclass R1 was enriched with

Hoshida's S1 (odds ratio (OR) = 12.9), Chiang's proliferation (OR = 4.5), and Sia's immune classes (OR = 14.4) (*p*-values < 0.00024)[16–18]. R2 was correlated with Hoshida's S2 (OR = 3.5) and Chiang's proliferation classes (OR = 2.2) (*p*-values < 0.0095). R3 was associated with Hoshida's S3 (OR = 6.0) and Chiang's CTNNB1 classes (OR = 3.4) (*p*-values < 0.00016). According to Hoshida et al., S1 reflects TGF-β-mediated activation of the WNT signaling pathway, S2 is proliferative tumor with MYC and AKT activation, and S3 is well-differentiated tumors[17].

### Therapeutic implications of the proteomic subclasses

Atezolizumab (anti-programmed death-ligand 1 (PD-L1)) plus bevacizumab (anti-vascular endothelial growth factor (VEGF)) showed superior efficacy to the multi-kinase inhibitor (MKI) sorafenib in treatment for unresectable HCC[4]. However, the response rate and duration of response to this treatment is still limited. It is therefore important to predict patient benefit and to understand the mechanism behind varying responses.

Tumor-infiltrating lymphocytes are an important factor for the therapeutic efficacy of immune checkpoint inhibitors. We examined five lymphocyte-expressed proteins (CD4, CD45, lymphocyte-specific protein tyrosine kinase (Lck), spleen associated tyrosine kinase (Syk), and ζ chain of T cell receptor associated protein kinase 70 (ZAP-70)) for their abundance in the proteomic subclasses (Fig. 5A). Expression levels of the five proteins were highest in the R1 subclass (*p*-values < $1 \times 10^{-10}$ by Wilcoxon rank sum test). mRNA-based signatures of immunological activity demonstrated a similar association with the R1 subclass (*p*-values < 0.0004). To validate this association, we performed immunohistochemistry of CD45, CD4, and CD8 on randomly selected 68 tumors in this cohort (Supplementary Fig. 5A). Density of positively stained immune cells was correlated with RPPA protein

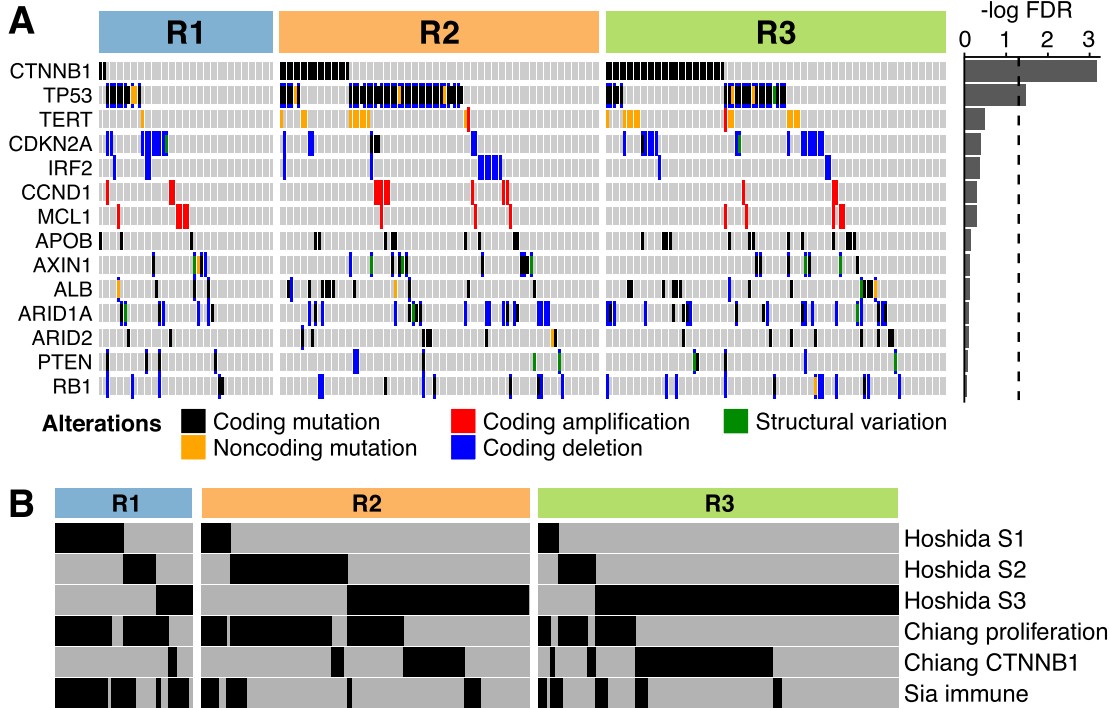

**Fig. 4 | Genomic and transcriptomic features of the proteomic subclasses.**
**A** Somatic driver alterations in the proteomic subclasses. Genes that were mutated in more than 10 tumors are shown. Bar chart on the right side shows strength of associations between alterations and subclasses, as measured by −log(FDR). FDR was computed by adjusting *p*-values of Fisher's exact test with the Benjamini−Hochberg method. Dotted line indicates FDR = 0.05. **B** Comparison between transcriptomic and proteomic subclasses. Only 204 HCCs where both RNA-Seq and RPPA data was available are shown.

expression levels for CD45 and CD4 (Supplementary Fig. 5B). R1 subclass had significantly higher density of CD45-positive cells than other subclasses (Supplementary Fig. 5C). R1 subclass of TCGA-LIHC also had significantly more lymphocyte infiltration than other subclasses (*p*-value < 0.01, Supplementary Fig. 4B). These results suggests that the R1 subclass is an immunologically "hot" tumor and may constitute a patient group that would selectively benefit from atezolizumab or other immune checkpoint inhibitors.

Vascular endothelial growth factor receptor 2 (VEGFR2) is a receptor of vascular endothelial growth factor (VEGF) and mediates tumor angiogenesis[19]. Many MKIs target VEGFR2. Protein abundance of VEGFR2 was highest in the R2 subclass (*p*-value < 0.0001 by Wilcoxon rank sum test) (Fig. 5B). This observation was replicated in TCGA-LIHC (*p*-value <0.01, Supplementary Fig. 4C). We also measured protein abundance of phospho-VEGFR2 using ELISA, and phospho-VEGFR2 also had a trend toward the highest expression in the R2 subclass (*p* = 0.062). The patients in the R2 subclass may selectively respond to bevacizumab or MKIs with selective inhibitory effect on VEGFR2 (e.g. lenvatinib) compared to sorafenib.

MKIs previously played a central role in the treatment of advanced HCC. Although MKIs are being replaced by atezolizumab plus bevacizumab as the first-line treatment, they will be still used as second- and third-line treatment of HCC[20]. We examined the expression levels of proteins targeted by MKIs sorafenib, lenvatinib, and regorafenib (Fig. 5B). Among the known targets of these MKIs, six proteins were available in our RPPA panel. Four of six proteins (VEGFR2, C-Raf, B-Raf, and C-Kit) were expressed at elevated levels in the R2 subclass (*p*-values <0.0001 by Wilcoxon rank sum test). Phosphorylated fibroblast growth factor receptor 4 (FGFR4), a major target of lenvatinib[21,22], did not correlate with the proteomic subclass (*p* = 0.37 by Kruskal−Wallis test). Because only 32 cases in this cohort were treated with MKI, disease control rate by MKI was not associated with the proteomic subclasses (R1, 50%; R2, 50%; R3, 13%; *p* = 0.18 by Fisher's

exact test). However, it is noteworthy that one case with a complete response to sorafenib belonged to the R2 subclass.

The R3 subclass contained lower levels of immune infiltrates, VEGF signaling, and MKI targets, suggesting a need for new effective approaches for this subclass. Importantly, the TSC/mTOR pathway had a high activity score in R3 (Fig. 3B). Concordantly, mTOR target sites of ribosomal protein S6 kinase B1 (S6K) and ULK1 have higher phosphorylation levels in R3 (Fig. 5C). High phospho-S6K in R3 was replicated in TCGA-LIHC (*p* < 0.001, Supplementary Fig. 4C). Known target sites of S6K also had higher phosphorylation levels in R3 (Fig. 5D). Moreover, hypoxia inducible factor 1 subunit α (HIF1A), which is upregulated by mTORC1[23], was enriched in R3 (Supplementary Fig. 6). Based on these results, we hypothesized that R3 tumors may respond to mTOR inhibitors or MKIs with higher inhibitory effect on the mTOR pathway. A mTOR inhibitor everolimus previously failed a clinical trial on HCC[24], but the result would be different if we could preselect likely responders.

### Efficacy of mTOR inhibitors on liver cancer cell lines

To test if mTOR activity predicts the response to pathway inhibitors, we explored public data of proteome and drug screening on liver cancer cell lines. RPPA data were obtained from Cancer Cell Line Encyclopedia (CCLE)[25], and drug screening data from the Cancer Therapeutics Response Portal (CTRP) project[26]. Growth inhibition by a mTOR inhibitor sirolimus on 20 liver cancer cell lines was strongly correlated with mTOR pathway activity (Supplementary Fig. 7A, B). Sirolimus strongly inhibited growth of the SNU398 cell line, which had the highest mTOR activity (IC$_{50}$: 2.0 nM). We also performed growth inhibition assays, applying another mTOR inhibitor temsirolimus to SNU398 and other liver cancer cell lines. Again, temsirolimus strongly inhibited growth of SNU398 (IC$_{50}$: 2.8 ± 0.7 nM) (Supplementary Fig. 7C). This was associated with a significant inhibition of mTOR downstream effectors, including phospho-p70S6K (Thr389)

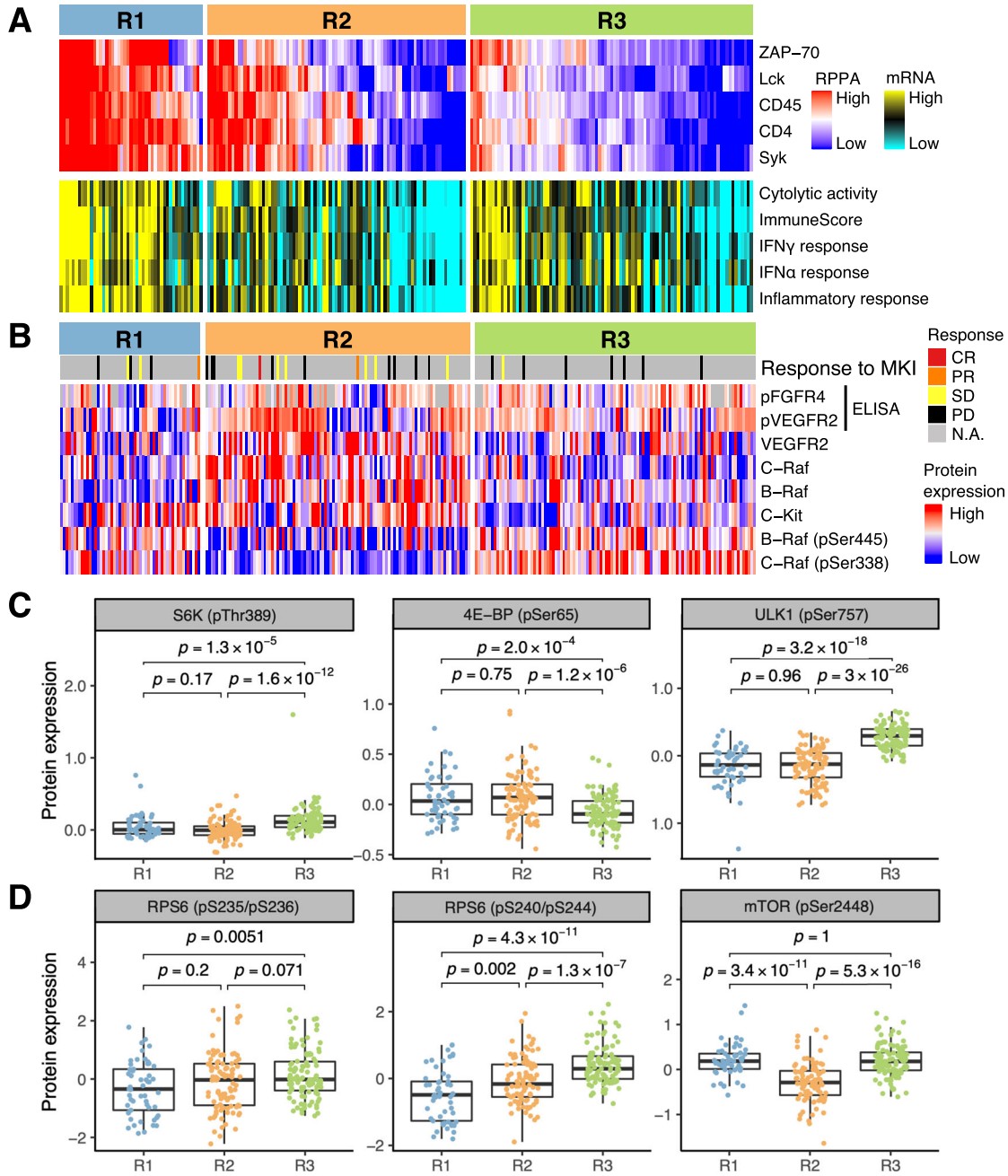

**Fig. 5 | Therapeutic implications of the proteomic subclasses. A** Immunity-related gene expression signatures in the proteomic subclasses. Protein expression levels of lymphocyte-associated genes (top) and mRNA signatures for immunological activity (bottom) are shown. **B** Expression levels of VEGFR2 and target proteins of MKIs. Expression levels were measured using ELISA (pFGFR4 and pVEGFR2) or RPPA (other six proteins). Therapeutic response of 32 patients to MKI (31 to sorafenib, 1 to lenvatinib) are also shown. CR complete response, PR partial response, SD stable disease, PD progressive disease. **C**, **D** Phosphorylation levels of **C** mTORC1 targets and **D** S6K targets. Sample sizes are $n = 53$ in R1, $n = 100$ in R2, and $n = 106$ in R3. Center of box shows the median. Lower and upper bounds of box are the first and third quartiles, respectively. Minima and maxima are the farthest values within 1.5 times the interquartile range from the bounds of box. Whiskers are drawn from the bounds of box to the extrema. ***$p < 0.001$; **$p < 0.01$; $^{NS}p \geq 0.05$. The $p$-values were computed using two-sided Wilcoxon rank sum test. Source data are provided as a Source Data file.

(Supplementary Fig. 7D, E). Interestingly, SNU398 has *CTNNB1* mutation, which is characteristic of the R3 subclass. The *CTNNB1* mutation and high mTOR activity is compatible with SNU398 originating from a R3 tumor. In contrast, JHH5 that has a *TP53* mutation[27], characteristic of the R2 subclass and low mTOR Signaling (Supplementary Fig. 7A-B) was resistant to temsirolimus (Supplementary Fig. 7C). On the other hand, activity of the HER2 inhibitor lapatinib was not associated with mTOR activity or response to temsirolimus (Supplementary Fig. 7C–E).

## Prognostic protein markers

Because immunohistochemistry is routinely performed in pathology laboratories, protein-based prognostic biomarkers have clinical utility. We examined 293 RPPA-measured proteins for their association with patient prognosis after surgery. Among the 293 proteins, 13 proteins were associated with poor overall survival, and 9 proteins were associated with good overall survival at FDR < 0.05 (Fig. 6A, B). Proteins associated with poor overall survival were dominated by cell cycle-related proteins (cyclin dependent kinase 1

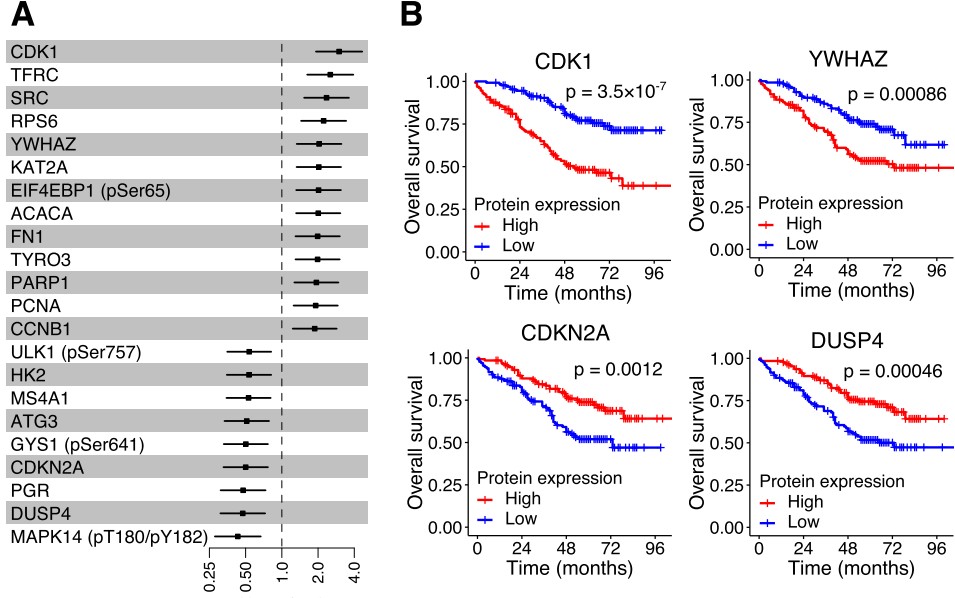

**Fig. 6 | Prognostic impact of protein expression levels in tumor. A** Hazard ratio for overall survival. For each of 293 proteins measured by RPPA, 259 patients were stratified into high- or low-expression groups based on the median expression levels, and their association with survival was computed using the Cox proportional hazard model. The p-values were adjusted for multiple hypothesis testing using the Benjamini–Hochberg method. Only proteins with FDR < 0.05 are shown.

Error bars show 95% confidence intervals. **B** Kaplan–Meier curves for overall survival and protein expression levels of CDK1, YWHAZ, CDKN2A, and DUSP4 in tumor. For each of the proteins, the number of patients were 129 in high-expression group and 130 in low-expression group. The p-values were computed by the log-rank test without adjustments for multiple comparisons. Source data are provided as a Source Data file.

(CDK1), SRC, poly(ADP-ribose) polymerase 1 (PARP1), proliferating cell nuclear antigen (PCNA), and cyclin B1 (CCNB1)). High expression of cyclin dependent kinase inhibitor 2 A (CDKN2A) was associated with good overall survival, consistently with its function as a tumor suppressor. No protein was associated with disease-free survival at FDR < 0.05.

Of the 13 proteins associated with a poor prognosis, five (SRC, tyrosine 3-monooxygenase/tryptophan 5-monooxygenase activation protein ζ (YWHAZ), lysine acetyltransferase 2A (KAT2A), fibronectin 1 (FN1), and TYRO3 protein tyrosine kinase (TYRO3)) and three (transferrin receptor (TFRC), ribosomal protein S6 (RPS6), and PCNA) were enriched in R1 and R2, respectively (Supplementary Fig. 8). None of proteins associated with a poor prognosis was upregulated in R3. In contrast, 7 of the 9 proteins associated with a good prognosis were more abundant in R3 (ULK1-pSer757, hexokinase 2 (HK2), membrane spanning 4-domains A1 (MS4A1), glycogen synthase 1 (GYS-pSer641), CDKN2A, PGR, and DUSP4) (Supplementary Fig. 9). Surprisingly based on the overall low association rate, eight proteins (eukaryotic translation initiation factor 4E binding protein 1 (EIF4EBP1), YWHAZ, TFRC, PARP1, DUSP4, acetyl-CoA carboxylase α (ACACA), RPS6, and CDKN2A) had significant positive correlations with SCNA copy numbers ($p < 0.05$).

**Genomic correlates of HCV.** One of unique characteristics of our cohort is frequent occurrence of HCV-positive tumors, which offers an opportunity to explore genomic correlates of HCV. We have reported characteristics of this cohort in several papers. We have reported whole-genome sequencing and mutation spectrum of 300 liver cancers[28]. In that paper, we found that *CTNNB1* mutation was significantly associated with HCV (OR = 1.55; $p = 0.0012$). *LRP1B* mutation also had association with HCV (OR = 1.66; $p = 0.0054$). In another paper, we analyzed RNA-seq data to characterize immune spectrum of this cohort[29]. We identified a subset of tumors where the Wnt/β-catenin signaling was activated, and HCV was significantly enriched in the subset (OR = 3.6; $p = 0.0007$).

To characterize HCV-specific signaling pathways, we first compared transcriptome profiles of HCV-positive and HCV-negative tumors using the gene set enrichment analysis. The analysis revealed that multiple pathways related to lipid metabolism were upregulated in HCV-positive tumors (Supplementary Fig. 10). We next compared RPPA pathway scores between HCV-positive and HCV-negative tumors. However, no pathway had statistically significant association with HCV probably because lipid metabolism pathway is not currently included in our RPPA pathways (Fig. 3B).

## Discussion

In this study, we performed large-scale proteomic profiling of primary liver cancer using RPPA. We analyzed 259 tumor tissues in a single study on the liver cancer proteome. Furthermore, our cohort is mainly constituted by HCV-positive patients in Japan. This expands the diversity of liver cancer proteome studies, which have mainly analyzed virus-negative Caucasian or HBV-positive Chinese patients[7,8,12]. Therefore, our proteome data should serve as a unique resource for liver cancer research.

Cluster analysis of our proteomic data revealed that liver cancers could be classified into three proteomic subclasses R1, R2, and R3. These subclasses had unique associations with clinical and genomic features of tumor and may facilitate patient stratification for systemic therapy. The R1 subclass expresses high levels of lymphocyte markers, indicating active infiltration of immune cells into tumor. Because T cell infiltration correlates with the response of tumor to immune checkpoint inhibitors[30], R1 may be more likely to respond to immune checkpoint blockade with inhibitors of programmed cell death 1 (PD1) or PD-L1 such as atezolizumab.

The R2 subclass has high AFP levels and is the most aggressive tumor among the three subclasses. R2 expresses significantly elevated levels of VEGFR2, suggesting activated VEGF signaling in this subclass. This is consistent with a previous finding that patients with high AFP levels benefit from anti-VEGFR2 antibody ramucirumab[31]. Therefore, R2 may have a higher chance of therapeutic response to bevacizumab,

which targets VEGF or other approaches that interfere with VEGF signaling.

The R3 tumors were smaller and less invasive, suggesting it may be amenable to locoregional therapy. However, if it progresses to advanced stages, R3 might be resistant to atezolizumab plus bevacizumab as it has low immune infiltration and VEGFR receptor expression. We found that R3 has the elevated activity of the mTOR signaling pathway. The EVOLVE-1 clinical trial failed to validate an mTOR inhibitor everolimus as the second line treatment for HCC after sorafenib[24]. However, subsets of HCC appear to respond to everolimus[32,33]. Where patient benefit to everolimus is selective to R3 subclass will require further exploration. Precise patient stratification may revive application of this drug to liver cancer therapy.

The R1 subclass was mostly immunologically "hot" tumors. Conversely, R2 and R3 contained many tumors with low immune gene expression (Fig. 5A). This is also evident from Fig. 4B, which shows that Sia's immune class was under-represented in R2 and R3. These "cold" tumors may not respond to immune checkpoint blockade.

Transcriptome analysis of tumor tissue has greatly improved the molecular analysis of HCC[16,17,34]. However, transcriptional analysis of HCC has not met with clinical deployment. Because immunohistochemistry of surgical specimen is routinely performed as a pathological examination, protein-based classification could fit better into clinical workflows. We identified proteins that were characteristically expressed in each subclass. These proteins may be proved to be useful biomarkers for classifying tumors into the proteomic subclasses.

One of limitations of this study is that tumor specimens used for RPPA were adjacent to but not identical to tumor areas used for RNA-seq. In a previous study, we found that different tumor nodules of the same liver often have divergent patterns of mRNA expression[35]. Therefore, pieces analyzed with RPPA may have substantially different gene expression patterns from pieces analyzed with RNA-seq.

In conclusion, this study presented the large dataset of liver cancer proteome and genome, which allows an exploration of molecular heterogeneity among primary liver cancers. The proteomic classification proposed in this study may guide precision treatment of patients with advanced liver cancer particularly for patients with HBV and HCV-associated liver cancer that have been under-represented in previous studies.

## Methods

### Clinical samples
Patients with primary liver cancer in Japan were enrolled in this study as a part of the International Cancer Genome Consortium (ICGC). All patients gave written informed consent for publishing their genome data and clinical information, following ICGC guidelines. Institutional review boards of RIKEN and the participating groups approved this study. Three hundred tumor specimens were obtained by surgical resection, freshly frozen, and analyzed using WGS and RNA-Seq[28]. After the sequencing analysis, 259 of 300 tumor specimens still had sufficient material to be analyzed using RPPA. Tumor specimens used for RPPA were adjacent to but not identical to tumor areas used for the sequencing analysis. Clinical information of the 259 patients are summarized in Table 1, and the details are available in Supplementary Data 1.

### RPPA profiling
RPPA was performed by the RPPA Core facility at the MD Anderson Cancer Center (MDACC), as described previously[5]. Briefly, tumor tissues were lysed with lysis buffer and Precellys homogenizers, and total protein concentration was measured using the bicinchoninic acid assay. Tumor lysates were serially diluted and spotted on nitrocellulose-coated slides. A set of 293 validated antibodies were individually applied to the slides (Set 142, the MDACC RPPA Core; Supplementary Data 2). The constantly updated list of validated

### Table 1 | Clinical information of liver cancer patients

| | |
|---|---|
| Number of patients | 259 |
| Age, median (Q1–Q3) | 69 (62–74) |
| Gender, male (%) | 195 (75) |
| Histology (%) | |
| HCC | 233 (90) |
| ICC | 20 (8) |
| cHCC-ICC | 6 (2) |
| Virus (%) | |
| HCV | 143 (55) |
| HBV | 62 (24) |
| NBNC | 50 (20) |
| HBV, HCV | 4 (2) |
| **Tumor size, ≥3 cm (%)** | 140 (54) |
| Vascular invasion (%) | 87 (34) |
| Stage (%) | |
| I | 39 (15) |
| II | 117 (45) |
| III | 79 (31) |
| IV | 24 (9) |
| **AFP, >200 ng/mL (%)** | 72 (28) |
| Platelet count, ≤100,000/µL | 58 (22) |
| Five-year overall survival (95% CI) | 0.63 (0.57–0.70) |
| Five-year disease-free survival (95% CI) | 0.37 (0.31–0.44) |
| Normal tissue for WGS (%) | |
| Lymphocyte | 256 (99) |
| Non-cancerous liver | 3 (1) |

antibodies are available at the RPPA Core Facility, MDACC (https://www.mdanderson.org/research/research-resources/core-facilities/functional-proteomics-rppa-core.html). Signal intensities were quantified and normalized to estimate protein expression levels. Log2 normalized expression levels were used throughout this study and is available in Supplementary Data 3. Hereafter, we refer this dataset as "RIKEN" dataset. Pathway scores were computed as described previously[5]. Member proteins of pathways are listed in Supplementary Data 4.

### Whole-genome sequencing
Whole-genome sequencing for 269 samples was performed as reported previously[28]. Genomic DNA was extracted from fresh-frozen tumor specimens. To facilitate mutation calling, normal DNA was also extracted from matched non-tumor specimens. Blood was used in most cases, and non-tumorous region of liver specimens were used otherwise. DNA libraries were constructed following the Illumina protocol. Sequencing was performed on HiSeq2000 or Genome Analyzer II platforms. Somatic mutation calls for single-nucleotide variants, insertions and deletions, copy number alterations, and structural variants were generated by the Pan-Cancer Analysis of Whole Genomes (PCAWG) project[36]. A list of pan-cancer driver genes was also obtained from PCAWG. Driver genes mutated in more than 10 liver tumors were analyzed for their enrichment in the proteomic subclasses using Fisher's exact test.

### RNA-seq
RNA-Seq experiments for 254 samples were performed as reported previously[28]. Total RNA was extracted from fresh-frozen tumor specimens. Poly(A) selection, cDNA synthesis, and library construction were performed according to the Illumina protocol. Sequencing was performed on HiSeq2000 platform. TopHat2 was used to map RNA-Seq

reads onto the reference human genome GRCh37. HTSeq was employed to count reads per gene, using GENCODE release 19 as a gene model. Fragments per kilobase of exon per million fragments mapped (FPKM) with upper quartile normalization was computed using in-house R script.

## Proteomic subclass

Proteins that have similar expression levels across samples will not be informative for tumor classification. Rather, they would amplify noises in measurements if their expression levels were scaled to have variance 1.0. Therefore, among 293 proteins quantified by RPPA, 171 proteins of low-expression variance (median absolute deviation < 0.3) were excluded. Expression levels of remaining proteins were scaled to have mean 0 and variance 1. To determine the number of clusters $k$, ConsensusClusterPlus (v1.46.0) was run with $k$ ranging from 2 through 10. For each run, 1000 iterations were performed using the Ward linkage and (1 − Pearson's correlation) distance. Consensus cumulative distribution function gradually rose as $k$ increased and did not have a clear-cut plateau (Supplementary Fig. 1A, B). However, at $k = 3$, clusters had consistent memberships while resampling and had distinct boundaries (Supplementary Fig. 1C, D). Larger $k$ values resulted in finer classification that retained the cluster structure of $k = 3$ (Supplementary Fig. 1E). Therefore, we employed $k = 3$ as the number of clusters in this dataset. Ten-fold cross-validation was performed with the R "caret" package using naive Bayes classifier and $k$-nearest neighbor classifier as prediction models.

## Validation dataset

RPPA data of The Cancer Genome Atlas Liver Hepatocellular Carcinoma (TCGA-LIHC) was used as a validation dataset. A normalized, batch-corrected expression matrix for 218 proteins in 184 tumors was downloaded from The Cancer Proteome Atlas Portal (https://tcpaportal.org/). Expression levels of 164 proteins that were shared between the RIKEN and TCGA datasets were extracted. Batch effects between the two datasets were corrected using ComBat[37]. To assign proteome subclasses to the TCGA datasets, supervised machine learning was employed. A naive Bayes classifier of the R "caret" package was trained with five-fold cross-validation of the RIKEN datasets. The input and output of the classifier were protein expression levels and a predicted subclass of each tumor, respectively. The classifier was subsequently used to assign proteome subclasses to 162 TCGA tumors whose clinical information was available in the paper of TCGA-LIHC[12].

## Transcriptomic subclass

Five gene sets for Chiang's classifications and three gene sets for Hoshida's classification of HCC were downloaded from MSigDB (v6.2)[16,17]. A gene set for immunological classification of HCC was acquired from the literature[18]. To assign transcriptomic subclass to each tumor, the NearestTemplatePrediction module of GenePattern server was used[38].

## Gene expression signatures

The following gene sets were downloaded from MSigDB: hallmark interferon-α response, hallmark interferon-γ response, and hallmark inflammatory response. Gene expression signatures were computed by feeding these gene sets and FPKM values to the ssGSEAProjection module of GenePattern server. Cytolytic activity was computed as a geometric mean between FPKM of *GZMA* and *PRF1*[39]. ImmuneScore was computed using the R package ESTIMATE[40].

## Immunohistochemistry

From 68 formalin-fixed, paraffin-embedded tissue samples, which were analyzed by RPPA in their frozen tissues, five 4 μm-thick sections were serially cut and mounted on pre-coated slides. CD45 assay was performed using DAKO Autostainer Link 48 (Agilent Technologies).

For antigen retrieval, the sections were heated at 97 °C for 20 min with Target Retrieval Solution, High pH (Agilent Technologies). For CD45, the tissue sections were stained by DAKO Envision FLEX-LCA (2B11 + PD7/26) kit. CD4 and CD8 assays were performed using the Ventana Benchmark XT system (Roche). For antigen retrieval, Cell Conditioning 1 (Roche) was poured onto the sections, which were then heated on a slide heater at 95 °C for 64 min. The tissue sections were incubated with prepared CD4 antibody (Ventana cat# 790-4423, 1:1) or prepared CD8 antibody (Ventana cat# 790-4460, 1:1).

## ELISA

The same tumor lysates as RPPA were analyzed for levels of phospho-VEGFR2 and phospho-FGFR4. PathScan® phospho-VEGFR2 antibody ELISA kit and PathScan® phospho-FGFR4 ELISA kit were used, respectively (Cell Signaling Technology cat#7824 and #69193 C). A standard curve was prepared by making serial dilution of the lysate of VEGF-Treated HUVEC Lysate (BioRad cat#171Z0010) for pVEGFR2 and FGF2-treated HepG2 lysate for pFGFR4. Liver cancer cell HepG2 cells were obtained from JCRB Cell Bank and were treated with 100 ng/mL of recombinant FGF2 (R&D System cat#233-FB-025/CF) for 5 min on 37 °C. After centrifugation, cells were lysed with RPPA buffer with TurboNuclease (Accelagen cat#N0103P) and disrupted in Bioruptor Level M for 10 sec. After centrifugation, supernatants were used to make a standardization curve for pFGFR4 ELISA. The phosphoprotein levels were normalized with total protein concentrations measured by the BCA assay. The normalized concentration data of pVEGFR2 and pFGFR4 are available as Supplementary Data 5.

## Cell line experiments

Five HCC cell lines, JHH1, JHH4, JHH5, JHH6, and SUN398, were obtained and from JCRB Cell Bank and ATCC and authenticated by them. All cell lines were grown in DMEM with 10% FBS at 37 °C with $CO_2$ after checking them as mycoplasma-free. Temsirolimus and lapatinib were purchased from selleckchem.com. Cell viability was determined using Trypan blue dye exclusion. Each cell line ($5.0 \times 10^3$ cells/ml $1 \times 10^4$ cells) was plated in a T75 flasco (15 cm) plate and incubated in medium with 10% FBS in the presence of temsirolimus, lapatinib or vehicle only. Viable cells were counted at day 4 using a Countess II automated cell counter (Thermo Fisher Scientific, Inc.). The growth inhibition rate was calculated as the ratio of the cell number in the presence of temsirolimus or lapatinib to that in the presence of vehicle only. Cells were seeded into a 96-well plate and grown overnight at 37 °C followed by treatment with temsirolimus or lapatinib at the indicated concentrations. Triplicate wells were used for each concentration. After an additional 72 h, CellTiter-Glo reagent was added into each well and incubation continued for 10 min at room temperature following the manufacturer's instructions (Promega). Luminescence signal intensity was measured by a microplate reader (TECAN).

## Signaling pathway analysis and western blot

Cells seeded in a T75 flasco (15 cm) plate were grown until 80% confluent. To determine mTOR signaling pathway or HER2 signaling pathway activity, cells were treated with temsirolimus or lapatinib for 4 days at 37 °C. Cells were washed twice with ice-cold PBS, and whole-cell lysates were prepared using lysis buffer (50 mmol/L Tris-HCl, pH 8.0, 150 mmol/L sodium chloride, 1.0% NP-40, 0.5% sodium deoxycholate, and 0.1% SDS) with sonication. Total protein concentration in the lysates was determined using a BCA Protein Assay Kit (Thermo Fisher Scientific Inc.). Protein lysates were subjected to SDS-PAGE and then transferred to a polyvinylidene difluoride membrane. After blocking using Bullet Blocking One for Western Blotting (NACALAI TESQUE, INC. Cat#13779-01) for 30 min at room temperature, the membranes were probed with phospho-p70S6 kinase (Cell Signaling Technology cat#9234, 1:500 dilution), phosphor-Akt (Ser473) (Cell Signaling Technology cat#4058, 1:500 dilution), Phospho-HER2/ErbB2

(Tyr1196) (Cell Signaling Technology cat#6942, 1:500 dilution), or Phospho-MEK1/2 (Ser217/221) (Cell Signaling Technology cat#9154, 1:500 dilution) at 4 °C overnight. Membranes were then washed with Blocking One (NACALAI TESQUE, INC. Cat#03953-95) and incubated with the secondary antibody to Anti-rabbit IgG HRP-linked Antibody (Cell Signaling Technology catalog#7074, 1:2000) and Precision Protein™ Strep Tactin-HRP Conjugate (catalog#161-0380, 1:50,000). at room temperature for 1 hour. After washing in Blocking One, the immunoblots were visualized using ECL detection reagents (NACALAI TESQUE, INC. cat#02230-30). GAPDH (D16H11) (Cell Signaling Technology cat#5174, 1:10,000) was used as a loading control. After photographing with the LAS-3000 (GE Healthcare JP), the area and intensity of the image files were measured using ImageJ, and the signal was calculated by integration.

### Reporting summary

Further information on research design is available in the Nature Research Reporting Summary linked to this article.

## Data availability

The raw whole-genome sequencing and RNA-seq data generated in this study have been deposited in NBDC under the accession numbers JGAD000228 and JGAD000229 (https://humandbs.biosciencedbc.jp/en/hum0158-v2). Somatic mutation call data for single-nucleotide variants, insertions and deletions, copy number alterations, and structural variants called by the Pan-Cancer Analysis of Whole Genomes (PCAWG) project[36] are available in ICGC DCC (https://dcc.icgc.org/releases). RPPA data of this study are available at The Cancer Proteome Atlas (TCPA) under the accession number TCPA00000009, as well as TCGA liver cancer (LIHC) dataset (https://tcpaportal.org/tcpa/). The results of ELISA for pVEGFR2 and pFGFR4 are available in Supplementary Data 5. CCLE data are available from https://depmap.org/portal/download/[25]. CTRP data are available from https://portals.broadinstitute.org/ctrp.v2.1/[26]. The remaining data are available within the Article, Supplementary Information, or Source Data file. Source data are provided with this paper.

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

## Acknowledgements

We would like to thank technical staff in RIKEN IMS and the RPPA Core facility at MDACC for their technical assistances. The super-computing resource 'SHIROKANE' was provided by the Human Genome Center, The University of Tokyo (http://supcom.hgc.jp/) and we acknowledge Ms. Hiroko Tanaka and other staff of SHIROKANE for their efforts on data management. This work was partly supported by JSPS KAKENHI Grant Number JP18H04049 awarded to H.N., and the Research Program on Hepatitis from the Japan Agency for Medical Research and Development (AMED) (21fk0310109h0005 and 21fk0210065h000221) to K.C., and National Cancer Institute grant R01-CA237327 and P50-CA217674 to J.S.L. The RPPA Core facility of MD Anderson Cancer Center was supported by the Core grant CA16672 from the National Cancer Institute.

## Author contributions

M.F., M.-J.M.C., S.S., H.L., S.S.N., and H.N. performed data analyses. D.R.S., A.O.-T., K.M., J.-S.L., Y.L., and G.B.M. performed sample processing and measurements of RPPA. S.S. and A.O.-T. performed ELISA and cell line experiments. A.O., R.M., H.A., M.U., S.H., H.Y., and K.C. collected clinical samples and clinical information. K.A. performed immunohistochemistry and its evaluation. M.F., G.B.M., S.S.N., and H.N. wrote the manuscript. H.N. and S.S.N. conceived the study and led the design of the experiments. H.N. contributed to the funding for this study.

## Competing interests

The authors declare no competing interests.

## Additional information

**Correspondence and requests** for materials should be addressed to Hidewaki Nakagawa.

