## [Peer Review File · Nature Communications]

Proteo-genomic Characterization of Virus-associated Liver Cancers Reveals Potential Subtypes and Therapeutic TargetsReviewers' Comments:

Reviewer #1:

Remarks to the Author:

Major point

1. The pathology of these enrolled samples, which include 233 HCC, 20 iCCA, and 6 HCC-iCCA mixed tumors, is quite complex and confusing. As these three types of tumors have different origins, their pathogen, mutation patterns and activated molecular pathways are significantly different. As a result, they are not suitable for cluster analysis when put together.

2. A number of high-throughput proteogenomics on HCC and iCCA have been carried out in the last three years. TMT-labeled proteomics is used in these studies to achieve higher-throughput and non-differential expression analysis. These studies included 110 early-stage HCC and 159 HBV-related HCC (Jiang, *Nature*, 2019; Gao, *Cell*, 2019), as well as 262 iCCA (Dong, *Cancer cell*, 2022). Therefore, this study lacks novelty and complementarity in terms of research design, sample size, and high-throughput technology selection. Also, it is difficult to claim that this is the largest sample cohort in the field of liver cancer proteome research.

3. The author chooses 300 proteins for RPPA and builds a prognostic model. What is the standard procedure for selecting these 300 proteins? The results of the study could be biased due to manual selection. The following is one possible strategy: the author narrows down the target protein based on previous positive results of higher-throughput screening such as TMT and DIA

4. The data for genomics and RNA expression profiling came from previously published studies (Fujimoto, *Nat. Genet*, 2016). The authors used RPPA proteomics to do more association analysis. The association analysis for these three omics, on the other hand, is quite limited and unconvincing. At the same time, the data quality isn't satisfied, and the protein-mRNA correlation isn't quite low (Figure 1, mean 0.25),

5. The proteogenomic characteristics of HCV patients may be the main spark of this study. However, the HCV-specific characteristics in the mutation spectrum, immune spectrum, and signaling pathways are insufficient in the current version, and may warrant further investigation.

Reviewer #3:

Remarks to the Author:

Genomic and proteomic technologies were used to characterize hepatocellular tumors from Japanese patients with virus associated liver cancer. This type of liver cancer is heterogeneous with variable treatment response rates. The goal of this study was to perform molecular profiling of a specific ethnic population to potentially elucidate molecular profiles for personalized medicine. Reverse phase protein array technology was used to profile protein expression. Three subsets of patients were identified based on genomic, transcriptomic, and proteomic profiles.

Overall, the results are highly relevant for a specific ethnic population. The proteomic methodologies are appropriate for inclusion in this multi-omic study. This study broadly supports proteomic and transcriptomic studies for personalized medicine.

The main drawback of the study is the data interpretation regarding immune signatures. The manuscript lacks histomorphology data to substantiate the presence/absence of tumor lymphocytes. The immune signatures are inferred from 5 protein levels, mRNA, and DNA methylation. Another drawback is the use of TCGA data as a validation set, rather than using another cohort of Japanese hepatocellular carcinoma samples and/or cross-validation machine learning methods. These concerns could be addressed in a revised manuscript.

Comments to authors:

1. Methods, page 6, lines 145-147: Please provide information in Table 1 regarding which cases used

normal adjacent tissue versus blood for extracting normal DNA. Were these samples randomly distributed between the 3 subgroups of patients identified by molecular profiling?

2. Methods: please provide a summary of the histology of these tumor samples. What was the tumor percentage in each sample? Was the same sample used for transcriptomic and proteomic analysis or were they from adjacent tumor areas? How similar were the different pieces of tissue that were used for the various analyses? See comments #8, 11, 13 and 15 below.

3. Methods, lines 165-166: "...171 proteins of low expression variance (median absolute deviation <0.3) were excluded." Please provide more explanation in the manuscript regarding why this large group of proteins were excluded from analysis. A 'variance in expression' compared to what?

4. Validation Data Set, lines 178-189: Validation sets should be samples from the same population used for the test set. Using samples from the TCGA add value in supporting the initial findings but should not be used to validate the results in this study. The TCGA does not contain adequate sample numbers of ethnic Japanese patients. Validation would be best using a cross-validation machine learning method or blinded analysis of a separate cohort of Japanese patients with hepatocellular carcinoma.

5. Results lines 265-267 and Figure 1: The data shown in figure 1 indicate that mRNA and protein abundances are slightly correlated, where as phosphoprotein and total protein abundances are not correlated. This substantiates previous literature. However, Figure 1 does not support the statement in lines 265-267 "However, of note, the correlation between RNA and protein levels represents a subset with high correlation coefficients supporting the utility of RPPA assay". The utility of RPPA or other proteomic methods is due to the fact that there is only slight correlation between transcriptomic and proteomic data and little to no correlation between transcriptomic and phosphoproteomic data. Please delete the sentence in lines 265-267.

6. Results lines 274-275: The weak correlation between phosphoprotein and total protein shown in Figure 1 and discussed in this Result section cannot be used to infer mechanistic control of cell signaling in liver cancer. Please delete the statement in lines 274-275.

7. Figure 2 and Results lines: 284-300: Please describe how you can determine true cis and trans interactions based only on the chromosome number that contain copy number alterations. Copy number alterations could be deletions, duplications, loss of heterozygosity, or copy neutral but cis and trans interactions reflect genomic interactions. Cis-acting factors affect gene expression only on the same chromosomal allele, while trans-acting factors act equally on both alleles. Cis-acting factors include enhancers, silencers, promoters, and epigenetic marks. Trans-acting factors include long noncoding RNAs. Experimental data is required to prove cis and trans interactions.

8. Results line 322: Subgroup R1 was described as "non-HCC histology". Please provide histomorphologic information regarding the tissue used in this study (see comment #2). How do you know that subgroup R1 tissue samples contained a high percentage of tumor? They could have contained a higher percentage of normal adjacent tissue compared to the other subgroups.

9. Results, line 345 and Figure 3B. Please list the hormones/hormone receptors that were included in the 2 hormone groups (hormone_a and hormone_b) shown in Figure 3B.

10. Results lines 368-370. Please provide reference citations for the different transcriptomic subclasses referenced from the literature (Hoshida, Chiang, and Sia). Define what "S1", "S2", and "S3" refer to in Hoshida's classes.

11. Figure 3 and Suppl Figure 3B: The percentage of TILs in each sample was not experimentally derived by histomorphology, which pinpoints the location of cell types. Furthermore, the abundance of TILs was estimated based on different methods in the test set compared to the validation set. In the test set, TILs were estimated based on 5 protein levels and mRNA signatures. In the validation set, using TCGA samples, TILs were "inferred from DNA methylation signatures". At best, the percentage of TILs can only be inferred in this manuscript without histomorphology and/or IHC data.

12. Figure 5A and line 430: Only subgroup R3 is noted in the text to contain a lower level of immune infiltrates. However, the samples on the right side of the heatmaps in R2 and R3 subgroups have low immune protein signaling and low cytolytic activity, immunoscore, and inflammatory response. Subgroups R2 and R3 appear to contain a subgroup of patient samples that have a low immune pathway signature. Please discuss the possibility that subgroups R2 and R3 contain a subset of samples with low immune expression. Consider further statistical analysis of the R2 and R3 subgroups

to investigate a low immune subset.

13. Figure 5A legend: The first sentence states "Immune infiltrates in the proteomic subclasses." The data shown does not support the presence or absence of immune infiltrates. Immune infiltrates should be shown by immunohistochemistry or histomorphology. See comment #15 also.

14. Figure 5C and D: define in the figure legend what the box blots and error bars represent. Are they mean, and SD, SEM, or 95% CI?

15. Line 502-503: The R1 subclass was found to have high expression of lymphocyte markers. The authors conclude that this indicates "...active infiltration of immune cells into tumor." Histology review and/or IHC must be done to determine the location of the immune cells.

16. Suppl Figure 5D, p70S6 western blot: Show the full-length western blots and MW marker for p70S6K. Explain the difference in single versus double bands in the blot images for the different cell lines. p70S6K is not documented to be a dimer and typically has a single band on western blots. Why does cell line JHH-6 have two bands whereas the other cell lines have only one band?

Minor comments:

17. Please provide both gene designation and protein name in the text. Both names are provided in Supp Table 2. Including both names in the text will facilitate the readers' association of genomic and proteomic data.

18. Line 399: Define "MTK".

19. Suppl Figure 4, 6, and 7 legends: change spelling of "Wilcoxson" to "Wilcoxon".

Reviewer #4:

Remarks to the Author:

In this study, the authors have performed RPPA proteomic profiling of ~300 proteins on 259 primary liver cancer tissues from Japanese patients. Based on these focused protein quantifications, the authors further performed a set of correlation and patient clustering analyses. Overall, the manuscript is clearly written and the RPPA data, generated on Asian ethnic backgrounds distinct from the TCGA cohort, is a useful resource. However, a potential limitation is the lack of new biological insights identified from many correlative analyses. I have a few suggestions for authors to consider.

Comment 1, Functional difference among proteins with high or low correlations with mRNA level. In Figure 1, the authors presented the difference in correlations between protein and mRNA levels and between protein and phosphorylation status. The extension of this analysis could potentially reveal novel biological insights:

1a, Are there any biological pathway or functional category differences between genes with high or low correlations in Fig. 1a?

1b, Are there any functional category differences between phosphorylation sites with high or low correlations in Fig. 1b?

1c, Comparing to other TCGA cohorts, are correlations consistent or cancer-type dependent? In another word, if a pair of protein and gene has a high (or low) correlation in this new cohort, does this pair have a high (or low) correlation in other TCGA cohorts?

Comment 2, Enumeration of genes with trans-interactions between SCNA and protein levels. Although in general, there is a lack of associations between SCNA and protein levels, there are still some chromosome arms, such as 4q, 5q, and 17p, showing enrichment of trans-interactions. Could the authors discuss what genes are in these regions and the potential cancer biology implications of these genes?

Comment 3, The clinical utility of R1, R2, and R3 classifications compared to data-integrative classifications using all genomics data modalities. The authors have defined R1-3 subtypes based on clustering RPPA data only and compared the consistency of proteomics clusters with other genomics measurements. However, an alternative approach is directly clustering using all data modalities, containing genomics alternations that can directly guide the choice of precision therapies.

Eventually, the goal of subtype clustering is to guide clinical decisions. Could the author justify the rationale for RPPA-only clusters? Alternatively, the authors could perform data-integrative clusters with RPPA data together with genomics data of frequently mutated genes profiled in the Foundation One or MSK IMPACT panels as such focused panel sequencing will become a norm in clinics with time. Then, the authors could evaluate the potential value of different clustering schemes to guide therapy decisions and suggest the most informative one for the community.

Comment 4, Further discussion of ethnic difference of prognostic protein markers. In the last section, the authors have analyzed protein markers associated with survival outcomes in different clusters. However, it will be more essential to see whether patients from different ethnic backgrounds in the TCGA cohort have the same correlations or distinct correlations. If protein markers with high associations with survival outcomes are different across distinct ethnic backgrounds, what are the potential genetic reasons or mechanistic explanations? For such a type of correlative analysis, implications on an essential biology question are critical.

REVIEWER COMMENTS

Reviewer #1, expertise in primary liver cancer subtypes and therapy, proteogenomics (Remarks to the Author):

Major point

1. The pathology of these enrolled samples, which include 233 HCC, 20 iCCA, and 6 HCC-iCCA mixed tumors, is quite complex and confusing. As these three types of tumors have different origins, their pathogen, mutation patterns and activated molecular pathways are significantly different. As a result, they are not suitable for cluster analysis when put together.

We agree with the reviewer that these could be different tumor types and that this could influence the clustering results. However, clustering across tumor types and tumor lineages does provide important information as long as the tumor type is known and thus this clustering approach does provide additional information for the manuscript. Pan-cancer cluster analysis of diverse tumor histology unveiled hidden similarities between different tumor types. For example, cluster analysis of TCGA showed that squamous cell cancers of several organs harbor shared molecular features^{1,2}. Even if the molecular pathways activated in some tumors are significantly different from others, it would not compromise the validity of cluster analysis; Instead, these tumors will simply form a distinct cluster.

Further several studies have proposed that some, if not all, of iCCAs and HCC-iCCAs originate from hepatocyte or transdifferentiate from HCC. Seehaver *et al.* reported that mutated hepatocytes can grow into either HCC or iCCA, depending on their microenvironment³. Hill *et al.* showed that hepatocytes with *KRAS* and *TP53* mutations can develop into iCCA⁴. Notch signaling plays a key role in development of iCCA from hepatocyte⁵⁻⁷. Genome analysis of HCC-iCCAs revealed that HCC components and iCCA components usually have monoclonal origin, suggesting plasticity of histology^{8,9}. This support the inclusion of all histological types of primary liver cancers in the subclassification approach.

Finally, exclusion of iCCA and HCC-iCCA did not substantially change our subclassification. To assess the impact of non-HCC histology on the proteomic subclasses, we excluded iCCA and HCC-iCCA and performed a clustering analysis of 233 HCC using ConsensusClusterPlus. Here, we temporarily named the resulting clusters as T1, T2, and T3. A table below shows the number of HCCs in the R and T clusters. The R clusters and T clusters were not independent each other ($p = 4.6 \times 10^{-68}$, the χ^2 test). Instead, in most cases (207 of 233 HCCs; 89%), one cluster corresponded to another: R1 to T1, R2 to T2, and R3 to T3.

	T1	T2	T3
R1	36	2	0
R2	18	77	0
R3	0	6	94

In term of the technical suitability of the sample types for cluster analysis, this is determined primarily by uniformity of sample processing and measurement procedure to decrease potential batch effects. Non-uniformity of experiments causes strong batch effects that hinder cluster analysis. In contrast, our data were generated in a highly uniform manner; Tumors were prospectively collected and processed following recommendations of the International Cancer Genome Consortium, and RPPA measurement was performed by the RPPA Core Facility of MD Anderson Cancer Center.

2. A number of high-throughput proteogenomics on HCC and iCCA have been carried out in the last three years. TMT-labeled proteomics is used in these studies to achieve higher-throughput and non-differential expression analysis. These studies included 110 early-stage HCC and 159 HBV-related HCC (Jiang, Nature, 2019; Gao, Cell, 2019), as well as 262 iCCA (Dong, Cancer cell, 2022). Therefore, this study lacks novelty and complementarity in terms of research design, sample size, and high-

throughput technology selection. Also, it is difficult to claim that this is the largest sample cohort in the field of liver cancer proteome research.

Although Dong *et al.* had 262 patients in their cohort, they performed proteome profiling on 214 tumors¹⁰ (Please see Figure S1A of Dong *et al.*). We therefore believe that our proteome data set still has the largest sample size (n = 259). However, out of respect for this reviewer's opinion, we toned down our claim and replaced the word "largest" with "large" in the manuscript. We also cited Dong *et al.* in the Introduction section.

Regarding research design, sample size, and high-throughput technology selection, there are multiple aspects of this study that are novel and complementary to the current literature.

Research design: Our research design meets the current standard of the field. Indeed, analyzed specimens were surgically resected and fresh frozen tumor tissues. In addition to proteomics profiling, we have performed whole genome sequencing and identified somatic mutations, copy number alterations, and structural variants. We have also performed RNA-seq and examined associations between genomic, transcriptomic, and proteomic data. All the data have been deposited to public repository to facilitate open and reproducible science. Three papers cited by this reviewer (Jiang, Gao, and Dong) also adopted a similar research design albeit with a smaller set of analytics. Therefore, we believe our research design is up to date.

Sample size: As we discussed above, our sample size (n = 259) appears to be the largest one in the field.

High-throughput technology selection: TMT proteomics has the ability to analyze a significant number of proteins. However, RPPA is also widely used and contemporary technology due to its increased sensitivity and in particular for its ability to quantify phosphoproteins especially regulatory phosphorylation sites (see PMID: 24719451 for a discussion). More than 60 papers including RPPA were published in each of the last three years (see figure below). This popularity may be partly due to its cost effectiveness, sensitivity and applicability to large sample numbers. Furthermore, compared with mass-spectrometry based proteomics technologies, RPPA has fewer batch effects increasing the reproducibility of the data. Taking advantage of these features, RPPA was actually applied to two large scale studies, TCGA and CCLE. Newly generated data of RPPA will thus have complementarity to these two important resources of cancer omics.

3. The author chooses 300 proteins for RPPA and builds a prognostic model. What is the standard procedure for selecting these 300 proteins? The results of the study could be biased due to manual selection. The following is one possible strategy: the author narrows down the target protein based on previous positive results of higher-throughput screening such as TMT and DIA

In the present study, we obtained assay results of 302 antibodies (RPPA Set 142) from RPPA core facility at MD Anderson Cancer Center. We selected antibodies based on the Probability value (QC score) higher than 0.8, which resulted in a set of 293 proteins for analysis. The QC score is an outcome

of a logistic regression model fitted with several QC predictors such as dynamic ranges and signal-to-background ratios of samples, and variations of 48 identical positive controls vertically spotted on RPPA slides¹¹. To minimize the antibody selection bias and cover broad signaling pathways, we set the following criteria: (1) interests from cancer research community (indicating their functions in cancer biology); (2) reviewed by RPPA Core at MD Anderson for accessibility; (3) processed for antibody validation in the RPPA Core at MD Anderson; and (4) confirmation of validation status of the antibodies in the RPPA technology. Our current antibody validation process is illustrated below.

Approach for Customer Antibody Validation (antibody validation for RPPA analysis)

4. The data for genomics and RNA expression profiling came from previously published studies (Fujimoto, *Nat. Genet.*, 2016). The authors used RPPA proteomics to do more association analysis. The association analysis for these three omics, on the other hand, is quite limited and unconvincing. At the same time, the data quality isn't satisfied, and the protein-mRNA correlation isn't quite low (Figure 1, mean 0.25),

We understand this reviewer's concern that our association analysis is limited. To dispel this concern, we performed additional association analysis. We compared somatic mutations of two driver genes, *CTNNB1* and *TP53*, with their protein expression levels. Mutations of *CTNNB1* were associated with elevated protein level ($p = 0.0033$) (Figure 2C). Mutations of *CTNNB1* disrupt phosphorylation sites of its exon 3, thereby preventing degradation of the protein¹². The elevated *CTNNB1* protein level is consistent with this mechanism. In contrast, mutations of *TP53* were significantly associated with decreased protein level ($p = 0.029$) (Figure 2D). This may be explained by frequent copy number loss of *TP53*. Indeed, in our data set, 88% of *TP53*-altered tumors had copy number loss of *TP53*.

Regarding data quality, the protein-mRNA correlation in our data set is low. However as indicated in multiple studies including our own RPPA and Mass Spec analysis, protein RNA correlations and in particular phosphoprotein RNA correlations are low. This emphasizes the importance of including proteomic analysis in studies of cancer cell types. Low protein-mRNA correlation does not necessarily indicate low data quality. Instead, the low correlation may simply reflect biological nature of analyzed tissues. Breen *et al.* analyzed 69 brain tissues and revealed that protein-mRNA correlation is inversely correlated with donor's age; neonatal brain has high correlation, and adult brain has low correlation¹³. Although their study used normal tissues, it clearly demonstrated that protein-mRNA correlation depends on biological nature of tissues. Nature of cancers varies country to country, having

different sex ratio, age, genetic background, and etiology. Therefore, direct comparison of correlation coefficients between different cohorts may not be valid.

Even if the low protein-mRNA correlation implies low data quality, it does not mean lower value of our data set as discussed below: Firstly, the protein-mRNA correlation is statistically significant ($p < 2.2 \times 10^{-16}$). This supports the data quality. Secondly, the data quality is complemented by large sample size of our data set. Based on the large sample size, our data set has high statistical power, and we were able to detect various associations between protein expression and clinical features as we reported in this study. Thirdly, our main result, proteomic subtype, is robust to measurement noise because it is defined by 122 protein markers. Measurement noise of individual proteins would cancel out each other and have minor effect on tumor classification. Fourthly, our data set is a unique proteome resource due to its predominantly HCV etiology. Because of these four reasons, we believe that our data set is worth sharing with the scientific community to promote omics research of liver cancer.

5. The proteogenomic characteristics of HCV patients may be the main spark of this study. However, the HCV-specific characteristics in the mutation spectrum, immune spectrum, and signaling pathways are insufficient in the current version, and may warrant further investigation.

We appreciate this reviewer's understanding on a strength of our cohort. We have reported characteristics of this cohort in several papers. We have reported whole genome sequencing and mutation spectrum of 300 liver cancers¹⁴. In that paper, we found that *CTNNB1* mutation was significantly associated with HCV (odds ratio (OR) = 1.55; $p = 0.0012$). *LRP1B* mutation was also associated with HCV (OR = 1.66; $p = 0.0054$). In another paper, we analyzed RNA-seq data to characterize immune spectrum of this cohort¹⁵. We identified a subset of tumors where the Wnt/ β -catenin signaling was activated, and HCV was significantly enriched in the subset (OR = 3.6; $p = 0.0007$).

To characterize HCV-specific signaling pathways, we first compared transcriptome profiles of HCV-positive and HCV-negative tumors using the gene set enrichment analysis. The analysis revealed that multiple pathways related to lipid metabolism were upregulated in HCV-positive tumors (**Supplementary Fig. 10**). We next compared RPPA pathway scores between HCV-positive and HCV-negative tumors. However, no pathway had statistically significant association with HCV probably because lipid metabolism pathway is not currently included in our RPPA pathways (**Fig. 3B**). We appended the above information to the tail of the Results section.

References for Reviewer #1

1. Hoadley, K. A. *et al.* Multiplatform analysis of 12 cancer types reveals molecular classification within and across tissues of origin. *Cell* **158**, 929–944 (2014).
2. Akbani, R. *et al.* A pan-cancer proteomic perspective on The Cancer Genome Atlas. *Nat. Commun.* **5**, 3887 (2014).
3. Seehawer, M. *et al.* Necroptosis microenvironment directs lineage commitment in liver cancer. *Nature* **562**, 69–75 (2018).
4. Hill, M. A. *et al.* Kras and Tp53 Mutations Cause Cholangiocyte- and Hepatocyte-Derived Cholangiocarcinoma. *Cancer Res.* **78**, 4445–4451 (2018).
5. Fan, B. *et al.* Cholangiocarcinomas can originate from hepatocytes in mice. *J. Clin. Invest.* **122**, 2911–2915 (2012).
6. Sekiya, S. & Suzuki, A. Intrahepatic cholangiocarcinoma can arise from Notch-mediated conversion of hepatocytes. *J. Clin. Invest.* **122**, 3914–8 (2012).
7. Wang, J. *et al.* Notch2 controls hepatocyte-derived cholangiocarcinoma formation in mice. *Oncogene* **37**, 3229–3242 (2018).
8. Joseph, N. M. *et al.* Genomic profiling of combined hepatocellular-cholangiocarcinoma reveals similar genetics to hepatocellular carcinoma. *J. Pathol.* **248**, 164–178 (2019).

9. Xue, R. *et al.* Genomic and Transcriptomic Profiling of Combined Hepatocellular and Intrahepatic Cholangiocarcinoma Reveals Distinct Molecular Subtypes. *Cancer Cell* **35**, 932-947.e8 (2019).
10. Dong, L. *et al.* Proteogenomic characterization identifies clinically relevant subgroups of intrahepatic cholangiocarcinoma. *Cancer Cell* **40**, 70-87.e15 (2022).
11. Ju, Z. *et al.* Development of a robust classifier for quality control of reverse-phase protein arrays. *Bioinformatics* **31**, 912–918 (2015).
12. Provost, E. *et al.* Functional correlates of mutations in beta-catenin exon 3 phosphorylation sites. *J. Biol. Chem.* **278**, 31781–9 (2003).
13. Breen, M. S. *et al.* Temporal proteomic profiling of postnatal human cortical development. *Transl. Psychiatry* **8**, 267 (2018).
14. Fujimoto, A. *et al.* Whole-genome mutational landscape and characterization of noncoding and structural mutations in liver cancer. *Nat. Genet.* **48**, 500–9 (2016).
15. Fujita, M. *et al.* Classification of primary liver cancer with immunosuppression mechanisms and correlation with genomic alterations. *EBioMedicine* **53**, 102659 (2020).

Reviewer #3, expertise in reverse-phase protein arrays (Remarks to the Author):

Genomic and proteomic technologies were used to characterize hepatocellular tumors from Japanese patients with virus associated liver cancer. This type of liver cancer is heterogeneous with variable treatment response rates. The goal of this study was to perform molecular profiling of a specific ethnic population to potentially elucidate molecular profiles for personalized medicine. Reverse phase protein array technology was used to profile protein expression. Three subsets of patients were identified based on genomic, transcriptomic, and proteomic profiles.

Overall, the results are highly relevant for a specific ethnic population. The proteomic methodologies are appropriate for inclusion in this multi-omic study. This study broadly supports proteomic and transcriptomic studies for personalized medicine.

The main drawback of the study is the data interpretation regarding immune signatures. The manuscript lacks histomorphology data to substantiate the presence/absence of tumor lymphocytes. The immune signatures are inferred from 5 protein levels, mRNA, and DNA methylation. Another drawback is the use of TCGA data as a validation set, rather than using another cohort of Japanese hepatocellular carcinoma samples and/or cross-validation machine learning methods. These concerns could be addressed in a revised manuscript.

Comments to authors:

1. Methods, page 6, lines 145-147: Please provide information in Table 1 regarding which cases used normal adjacent tissue versus blood for extracting normal DNA. Were these samples randomly distributed between the 3 subgroups of patients identified by molecular profiling?

We used lymphocytes in 256 cases and noncancerous livers in 3 cases for extracting normal DNA. We added this information to Table 1. The 3 cases were found evenly in R1, R2, and R3 subclasses.

2. Methods: please provide a summary of the histology of these tumor samples. What was the tumor percentage in each sample? Was the same sample used for transcriptomic and proteomic analysis or were they from adjacent tumor areas? How similar were the different pieces of tissue that were used for the various analyses? See comments #8, 11, 13 and 15 below.

Table 1 includes a summary of histology of tumors. The number of HCC, ICC, cHCC-ICC were 233 cases, 20 cases, and 6 cases, respectively. We appended tumor purity to Supplementary Table 1. This tumor purity was estimated by WGS data analysis (Nature, 2020, 578:82–93) and available for 251 of 259 tumors. The median of tumor purity was 66%.

Transcriptomic and proteomic analyses used adjacent tumor areas. We added this description to the Methods section. Regarding similarity of different pieces of tumor: in a previous study, we compared mRNA expression patterns between different tumor nodules of the same liver (Furuta et al., J Hepatol. 2017, 66:363-373). The study showed that the tumor nodules often have divergent mRNA expression patterns. Therefore, pieces analyzed with RPPA and pieces analyzed with RNAseq may have substantial differences in their gene expression patterns, although they were located closely with the tumor. We describe this as one of limitations of this study in the Discussion section.

3. Methods, lines 165-166: "...171 proteins of low expression variance (median absolute deviation <0.3) were excluded." Please provide more explanation in the manuscript regarding why this large group of proteins were excluded from analysis. A 'variance in expression' compared to what?

Proteins that have similar expression levels across samples will not be informative for tumor classification. Rather, they would amplify noises in measurements if their expression levels were scaled to have variance 1.0. This is the reason why we excluded low-variance proteins from tumor classification. We have added this explanation to the Method section. A similar step has been commonly used by previous studies of mRNA-based tumor classifications. For each protein of each sample, the expression level was compared to the median expression level of the protein in the cohort,

and their absolute deviations from the median were computed. The median of the absolute deviations was used as an index of variation level of the protein.

4. Validation Data Set, lines 178-189: Validation sets should be samples from the same population used for the test set. Using samples from the TCGA add value in supporting the initial findings but should not be used to validate the results in this study. The TCGA does not contain adequate sample numbers of ethnic Japanese patients. Validation would be best using a cross-validation machine learning method or blinded analysis of a separate cohort of Japanese patients with hepatocellular carcinoma.

Following the suggestion of this reviewer, we used a cross-validation machine learning method. We applied 10-fold cross validation to the protein expression matrix of Japanese cohort (Fig. 3A; 122 proteins \times 259 tumors). Tumors were randomly split into 10 subsets. Nine of them were used for training, the remaining one was used for testing, and this process was repeated 10 times to compute the average prediction accuracy. Naive Bayes classifier was trained to predict subclasses R1, R2, and R3 from protein expression levels. Its average prediction accuracy was high (91.5%). When k -nearest neighbor classifier was used as a prediction model, it also showed high prediction accuracy (91.1%; $k = 3$). These cross-validation analysis confirmed the robustness of our classification. We have added this analysis to the Results section.

5. Results lines 265-267 and Figure 1: The data shown in figure 1 indicate that mRNA and protein abundances are slightly correlated, where as phosphoprotein and total protein abundances are not correlated. This substantiates previous literature. However, Figure 1 does not support the statement in lines 265-267 “However, of note, the correlation between RNA and protein levels represents a subset with high correlation coefficients supporting the utility of RPPA assay”. The utility of RPPA or other proteomic methods is due to the fact that there is only slight correlation between transcriptomic and proteomic data and little to no correlation between transcriptomic and phosphoproteomic data. Please delete the sentence in lines 265-267.

We agree the reviewer’s opinion. We deleted the sentence.

6. Results lines 274-275: The weak correlation between phosphoprotein and total protein shown in Figure 1 and discussed in this Result section cannot be used to infer mechanistic control of cell signaling in liver cancer. Please delete the statement in lines 274-275.

We admit the sentence was too speculative. We deleted the sentence.

7. Figure 2 and Results lines: 284-300: Please describe how you can determine true cis and trans interactions based only on the chromosome number that contain copy number alterations. Copy number alterations could be deletions, duplications, loss of heterozygosity, or copy neutral but cis and trans interactions reflect genomic interactions. Cis-acting factors affect gene expression only on the same chromosomal allele, while trans-acting factors act equally on both alleles. Cis-acting factors include enhancers, silencers, promoters, and epigenetic marks. Trans-acting factors include long noncoding RNAs. Experimental data is required to prove cis and trans interactions.

We have now clarified the terminology as requested. In our revised manuscript, we replaced the words “cis” and “trans” with “local” and “distant”, respectively.

8. Results line 322: Subgroup R1 was described as “non-HCC histology”. Please provide histomorphologic information regarding the tissue used in this study (see comment #2). How do you know that subgroup R1 tissue samples contained a high percentage of tumor? They could have contained a higher percentage of normal adjacent tissue compared to the other subgroups.

We apologize for a confusing description. In the previous version of manuscript, we wrote “*The R1*

subclass was ... more likely to ... have non-HCC histology (ICC and cHCC-ICC; 28% vs 5%; $p = 1.0 \times 10^{-5}$). Here, the numbers “28% vs 5%” did not mean tumor cell percentage. They are the proportion of tumor samples that were histologically classified as intrahepatic cholangiocarcinoma (ICC) and combined hepatocellular carcinoma-intrahepatic cholangiocarcinoma (cHCC-ICC) in the subclasses. For example, among 53 tumors of R1, 38, 11, and 4 tumors were HCC, ICC, and cHCC-ICC, respectively. Therefore, the percentage of non-HCC histology was $15 / 53 = 28\%$. We revised the description to clarify this point. We also provided the histological information of each sample in Supplementary Table 1.

9. Results, line 345 and Figure 3B. Please list the hormones/hormone receptors that were included in the 2 hormone groups (hormone_a and hormone_b) shown in Figure 3B.

We created Supplementary Table 3, which lists member proteins of the hormone_a and hormone_b pathway.

10. Results lines 368-370. Please provide reference citations for the different transcriptomic subclasses referenced from the literature (Hoshida, Chiang, and Sia). Define what “S1”, “S2”, and “S3” refer to in Hoshida’s classes.

We added reference citations for Hoshida, Chiang, and Sia to the Results section. According to Hoshida et al., S1 reflects TGF- β -mediated activation of the WNT signaling pathway, S2 is proliferative tumor with MYC and AKT activation, and S3 is well-differentiated tumors. We also added this explanation to the main text.

11. Figure 3 and Suppl Figure 3B: The percentage of TILs in each sample was not experimentally derived by histomorphology, which pinpoints the location of cell types. Furthermore, the abundance of TILs was estimated based on different methods in the test set compared to the validation set. In the test set, TILs were estimated based on 5 protein levels and mRNA signatures. In the validation set, using TCGA samples, TILs were “inferred from DNA methylation signatures”. At best, the percentage of TILs can only be inferred in this manuscript without histomorphology and/or IHC data.

For the RIKEN dataset, we randomly selected 68 tumors (23 R1, 22 R2, and 23 R3 tumors) and performed immunohistochemistry of CD45, CD4, and CD8. Density of positively stained immune cells was evaluated in a semi-quantitative manner by an experienced pathologist (K.A.). Density of positively stained cells was correlated with RPPA protein expression levels for CD45 and CD4 (Supplementary Fig. 5B). CD8 antibody was not included in the antibody set of RPPA.

R1 subclass had significantly higher density of CD45-positive cells than other subclasses (Supplementary Figure 5). For the TCGA dataset, we replaced Supplementary Figure 4B with H&E-based measurement of TIL.

12. Figure 5A and line 430: Only subgroup R3 is noted in the text to contain a lower level of immune infiltrates. However, the samples on the right side of the heatmaps in R2 and R3 subgroups have low immune protein signaling and low cytolytic activity, immunoscore, and inflammatory response. Subgroups R2 and R3 appear to contain a subgroup of patient samples that have a low immune pathway signature. Please discuss the possibility that subgroups R2 and R3 contain a subset of samples with low immune expression. Consider further statistical analysis of the R2 and R3 subgroups to investigate a low immune subset.

We agree this reviewer’s opinion that R2 and R3 contain immunologically “cold” tumors. As we have reported in the original version of manuscript, the R1 subclass was associated with immunologically “hot” tumors. Conversely, R2 and R3 contained many tumors with low immune gene expression (Fig. 5A). This is also evident from Figure 4B, which shows that Sia’s immune class was underrepresented in R2 and R3. We added these points at the end of Discussion section. Regarding subset analysis: we split R2 into “hot R2” and “cold R2” by intersecting it with Sia’s immune class. We compared “hot R2” and “cold R2” for their statistical association with following clinical features: age, sex, tumor size,

vascular invasion, stage, platelet count, AFP, HBV, and HCV. Only age had statistically significant difference ($p < 0.05$). We performed a similar analysis on R3, but no clinical features had significant difference. Because of the limited significant associations, we did not add this subset analysis to the revised manuscript.

13. Figure 5A legend: The first sentence states “Immune infiltrates in the proteomic subclasses.” The data shown does not support the presence or absence of immune infiltrates. Immune infiltrates should be shown by immunohistochemistry or histomorphology. See comment #15 also.

We corrected Figure 5A legend as “Immunity-related gene expression signatures”.

14. Figure 5C and D: define in the figure legend what the box blots and error bars represent. Are they mean, and SD, SEM, or 95% CI?

Boxes and error bars in the figure represent quartiles and inter-quartile range. We added this description to the figure legend.

15. Line 502-503: The R1 subclass was found to have high expression of lymphocyte markers. The authors conclude that this indicates “...active infiltration of immune cells into tumor.” Histology review and/or IHC must be done to determine the location of the immune cells.

Our claim is now supported by immunohistochemistry (Supplementary Fig. 5). For the RIKEN dataset, we randomly selected 68 tumors (23 R1, 22 R2, and 23 R3 tumors) and performed immunohistochemistry of CD45, CD4, and CD8. Density of positively stained immune cells was evaluated in a semi-quantitative manner by an experienced pathologist (K.A.). Density of positively stained cells was correlated with RPPA protein expression levels for CD45 and CD4 (Supplementary Fig. 5B). CD8 antibody was not included in the antibody set of RPPA.

16. Suppl Figure 5D, p70S6 western blot: Show the full-length western blots and MW marker for p70S6K. Explain the difference in single versus double bands in the blot images for the different cell lines. p70S6K is not documented to be a dimer and typically has a single band on western blots. Why does cell line JHH-6 have two bands whereas the other cell lines have only one band?

We wish to thank the reviewer for this comment. We performed western blot using antibodies for Phospho-p70S6 Kinase (Thr389) (108D2) Rabbit mAb #9234 (<https://www.cellsignal.jp/products/primary-antibodies/phospho-p70-s6-kinase-thr389-108d2-rabbit-mab/9234>), The results showed a double band similar to that noted in the product information. We believe that the double band was detected because p70S6K has 5 isoforms, P85s6K and P70s6K, which is deleted by 23 residues from the N-terminal, were detected.

Minor comments:

17. Please provide both gene designation and protein name in the text. Both names are provided in Supp Table 2. Including both names in the text will facilitate the readers' association of genomic and proteomic data.

We described both protein names and gene names throughout the manuscript.

18. Line 399: Define "MTK".

This was a typo of "multi-kinase inhibitor (MKI)". It is now corrected.

19. Suppl Figure 4, 6, and 7 legends: change spelling of "Wilcoxson" to "Wilcoxon".

We have corrected this spelling error.

Reviewer #4, expertise in multi-omics/bioinformatics (Remarks to the Author):

In this study, the authors have performed RPPA proteomic profiling of ~300 proteins on 259 primary liver cancer tissues from Japanese patients. Based on these focused protein quantifications, the authors further performed a set of correlation and patient clustering analyses. Overall, the manuscript is clearly written and the RPPA data, generated on Asian ethnic backgrounds distinct from the TCGA cohort, is a useful resource. However, a potential limitation is the lack of new biological insights identified from many correlative analyses. I have a few suggestions for authors to consider.

Comment 1, Functional difference among proteins with high or low correlations with mRNA level. In Figure 1, the authors presented the difference in correlations between protein and mRNA levels and between protein and phosphorylation status. The extension of this analysis could potentially reveal novel biological insights:

1a, Are there any biological pathway or functional category differences between genes with high or low correlations in Fig. 1a?

Following this reviewer's comment, we performed three functional enrichment analyses: (1) GO enrichment analysis for proteins with the top 25% correlations; (2) GO analysis for the bottom 25%; (3) GSEA Pre-ranked analysis weighted by correlation coefficients. Unfortunately, none of these analyses had statistically significant hits.

1b, Are there any functional category differences between phosphorylation sites with high or low correlations in Fig. 1b?

The same analyses were also applied to phosphoproteins, but there were no significant pathways.

1c, Comparing to other TCGA cohorts, are correlations consistent or cancer-type dependent? In another word, if a pair of protein and gene has a high (or low) correlation in this new cohort, does this pair have a high (or low) correlation in other TCGA cohorts?

We examined "correlation of correlations" between our RIKEN cohort and 11 TCGA cohorts. To do so, we first computed correlation coefficients of mRNA-protein pairs in the RIKEN cohorts. We next computed mRNA-protein correlations in TCGA cohorts. We then compared the correlation coefficients of the same mRNA-protein pairs between RIKEN and TCGA. We made a new Supplementary Figure 2 to show the result. All 11 TCGA cohorts had significantly positive "correlation of correlations". TCGA liver cancer cohort (LIHC) had the highest correlation of correlations (Spearman's correlation $r=0.62$). This result shows that correlations are partly dependent and partly independent on cancer types.

Comment 2, Enumeration of genes with trans-interactions between SCNA and protein levels. Although in general, there is a lack of associations between SCNA and protein levels, there are still some chromosome arms, such as 4q, 5q, and 17p, showing enrichment of trans-interactions. Could the authors discuss what genes are in these regions and the potential cancer biology implications of these genes?

Because arm-level SCNA affect hundreds of genes, it is difficult to pinpoint a single causative gene of trans-interactions. However, we can list oncogenes or tumor suppressor genes in these chromosome arms. Loss of 17p was observed in 49% of cases, and the chromosome arm has a tumor suppressor gene *TP53*. Loss of 4q was found in 35% of cases, and the arm contains interferon regulatory factor 2 (*IRF2*) gene. A previous study reported that *IRF2* is a tumor suppressor gene as its inactivation impairs TP53 function in HCC (Guichard et al., Nature Genetics, 44:694–698 (2012)). Gain of 5q was found in 25% of cases, and the arm contains *FGFR4* gene. We added these gene names to the Results section. It is tempting to speculate that SCNA of these genes caused trans-interaction in proteome. However, future cell line experiments will be necessary to prove such hypotheses.

Comment 3, The clinical utility of R1, R2, and R3 classifications compared to data-integrative classifications using all genomics data modalities. The authors have defined R1-3 subtypes based on clustering RPPA data only and compared the consistency of proteomics clusters with other genomics measurements. However, an alternative approach is directly clustering using all data modalities, containing genomics alternations that can directly guide the choice of precision therapies. Eventually, the goal of subtype clustering is to guide clinical decisions. Could the author justify the rationale for RPPA-only clusters? Alternatively, the authors could perform data-integrative clusters with RPPA data together with genomics data of frequently mutated genes profiled in the Foundation One or MSK IMPACT panels as such focused panel sequencing will become a norm in clinics with time. Then, the authors could evaluate the potential value of different clustering schemes to guide therapy decisions and suggest the most informative one for the community.

Antibody-based immunohistochemistry (IHC) assay is widely used for pathological diagnosis of cancers not only in developed countries and but also in developing countries. IHC is applicable to both new and archived specimens and shows qualitative expression levels of proteins with high molecular specificity, even at the level of isoforms and phospho-isoforms. Because our RPPA-only clusters are based on protein expression levels assayed by antibodies, we expect that it will have high affinity and concordance with IHC and clinical practice. If cluster markers are selected appropriately, IHC of two or three cluster markers may be able to map our proteomic classification onto clinical tumor samples, which may guide clinical decision making and also give insights into prognosis of patients.

Comment 4, Further discussion of ethnic difference of prognostic protein markers. In the last section, the authors have analyzed protein markers associated with survival outcomes in different clusters. However, it will be more essential to see whether patients from different ethnic backgrounds in the TCGA cohort have the same correlations or distinct correlations. If protein markers with high associations with survival outcomes are different across distinct ethnic backgrounds, what are the potential genetic reasons or mechanistic explanations? For such a type of correlative analysis, implications on an essential biology question are critical.

In the TCGA liver cancer cohort (TCGA-LIHC), we tested expression levels of 218 proteins for their associations with overall survival of patients. We analyzed only 162 tumors where both RPPA measurements and clinical metadata were available. Unfortunately, no protein had significant association with overall survival at $FDR < 0.05$. This low statistical power in TCGA could be explained by differences in sample sizes (162 tumors in TCGA; 259 tumors in RIKEN) and follow-up time (median 20.4 months in TCGA; 50.0 months in RIKEN).

Reviewers' Comments:

Reviewer #1:

Remarks to the Author:

The authors have addressed the comments that were raised. I'm satisfied with the revised manuscript and have no additional comments.

Reviewer #3:

Remarks to the Author:

The revised manuscript adequately addresses the reviewer's comments. Thank you for the detailed responses.

Reviewer #4:

Remarks to the Author:

In the revision, the authors have addressed my previous comments and concerns. I think this work is suitable for publication, and the data will be useful to the cancer genomics community.

REVIEWERS' COMMENTS

Reviewer #1 (Remarks to the Author):

The authors have addressed the comments that were raised. I'm satisfied with the revised manuscript and have no additional comments.

> Thank you.

Reviewer #3 (Remarks to the Author):

The revised manuscript adequately addresses the reviewer's comments. Thank you for the detailed responses.

> Thank you.

Reviewer #4 (Remarks to the Author):

In the revision, the authors have addressed my previous comments and concerns. I think this work is suitable for publication, and the data will be useful to the cancer genomics community.

> Thank you. Our data is deposited into the public database and available.